evolution/taxonomy and systematics/ palaeontology

Late Cretaceous, Maastrichtian, Maevarano Formation, *Testudines*, *Pleurodira*, Sahonachelyidae

**Author for correspondence:**
Walter G. Joyce
e-mail: walter.joyce@unifr.ch

# A new pelomedusoid turtle, *Sahonachelys mailakavava*, from the Late Cretaceous of Madagascar provides evidence for convergent evolution of specialized suction feeding among pleurodires

Walter G. Joyce[1], Yann Rollot[1], Serjoscha W. Evers[1], Tyler R. Lyson[2], Lydia J. Rahantarisoa[3] and David W. Krause[2,4]

[1]Departement für Geowissenschaften, Universität Freiburg, Fribourg, Switzerland
[2]Department of Earth Sciences, Denver Museum of Nature & Science, Denver, CO, USA
[3]Département de Sciences de la Terre et de l'Environnement, Université d'Antananarivo, Antananarivo, Madagascar
[4]Department of Anatomical Sciences, Stony Brook University, Stony Brook, NY, USA

(iD) WGJ, 0000-0003-4726-2449; YR, 0000-0002-2020-9456; SWE, 0000-0002-2393-5621; TRL, 0000-0003-4391-9044; LJR, 0000-0003-3216-6630; DWK, 0000-0001-7860-6828

The Maevarano Formation in northwestern Madagascar has yielded a series of exceptional fossils over the course of the last three decades that provide important insights into the evolution of insular ecosystems during the latest Cretaceous (Maastrichtian). We here describe a new genus and species of pelomedusoid turtle from this formation, *Sahonachelys mailakavava*, based on a nearly complete skeleton. A phylogenetic analysis suggests close affinities of *Sahonachelys mailakavava* with the coeval Madagascan *Sokatra antitra*. These two taxa are the only known representatives of the newly recognized clade Sahonachelyidae, which is sister to the speciose clade formed by Bothremydidae and Podocnemidoidae. A close relationship with coeval Indian turtles of the clade Kurmademydini is notably absent. A functional assessment suggests that *Sahonachelys*

mailakavava was a specialized suction feeder that preyed upon small-bodied invertebrates and vertebrates. This is a unique feeding strategy among crown pelomedusoids that is convergent upon that documented in numerous other clades of turtles and that highlights the distinct evolutionary pathways taken by Madagascan vertebrates.

## 1. Introduction

Extant pelomedusoid turtles are found across much of Africa, Madagascar and South America. Although the clade diverged as early as the Early Cretaceous, approximately the same time as chelids, trionychids and durocryptodires [1–3], today they comprise only about 10 per cent of species diversity and 5 per cent of generic diversity [4]. Over the course of the last 25 years, however, an astounding diversity of fossil forms has been documented from Cretaceous to Palaeocene strata (e.g. [5–21]). These fossils not only expand the range of pelomedusoids to Arabia, the Caribbean, Europe, India and North America but also document an incredible array of diversity and disparity within the group. Although a formal ecomorphological analysis is still outstanding for the entire clade, fossil and extant pelomedusoids generally possess high skulls with well-developed triturating surfaces that suggest herbivorous, omnivorous and durophagous diets [8,22].

In 2015, a pelomedusoid turtle skeleton was discovered while removing overburden from a productive archosaur site within the Maevarano Formation of northwestern Madagascar. The specimen is not only unusual for its fragility and completeness but also in displaying numerous morphological adaptations consistent with specialized suction feeding. The purpose of this contribution is to describe this specimen as a new genus and species of turtle, *Sahonachelys mailakavava*, to investigate its phylogenetic relationships, and to assess its palaeoecology.

## 2. Geological setting

The holotype and only known specimen of *Sahonachelys mailakavava* gen. et sp. nov., Université d'Antananarivo (UA) 10581, was discovered in June 2015 at locality MAD05-38 in the Berivotra Study Area of the Mahajanga Basin, northwestern Madagascar, while removing overburden less than 1 m above a layer rich in dinosaur and crocodyliform fossils (figure 1). Locality MAD05-38 occurs in the Anembalemba Member of the Maevarano Formation [23]. Locality coordinates are on file at the Denver Museum of Nature & Science and the Université d'Antananarivo and are available to qualified investigators.

The Maevarano Formation is of Maastrichtian (latest Cretaceous) age (see summary of geochronological data in [24]). The vertebrate assemblage from the formation includes dipnoan fishes (1 sp.), actinopterygian fishes (10 spp.), frogs (2 spp.), turtles (5 spp., including *S. mailakavava*), snakes (6 spp.), non-ophidian squamates (1 sp.), crocodyliforms (6 spp.), titanosaurian sauropod dinosaurs (2 spp.), non-avian theropod dinosaurs (3 spp.), avialans (6 spp.) and mammals (7 spp.), for a total of 51 currently known species [25]. The formation is particularly well known for yielding exquisitely preserved small-bodied vertebrates, including the frog *Beelzebufo ampinga* [26], the cordylid lizard *Konkasaurus mahalana* [27], the notosuchian *Araripesuchus tsangatsangana* [28], the paravian theropod *Rahonavis ostromi* [29,30], the enantiornithine theropod *Falcatakely forsterae* [31], and the gondwanatherian mammals *Vintana sertichi* [32] and *Adalatherium hui* [25]. In addition to fragmentary material that may represent additional species [25], the Maevarano Formation has also yielded cranial material of the bothremydid turtle *Kinkonychelys rogersi* [33], the basal pelomedusoid *Sokatra antitra* [34] and an unnamed podocnemidoid [35].

The Maevarano Formation was deposited at a time when northwestern Madagascar had a highly seasonal (pronounced dry and wet seasons), semi-arid climate and was positioned at approximately 30° S latitude (15° farther south than today), in the subtropical desert belt [24,25]. Madagascar was isolated in the Indian Ocean, having been separated from the African mainland before 165 Ma, from Antarctica–Australia at approximately 124 Ma, and the Indian subcontinent at approximately 88 Ma, approximately 20 Myr earlier than when the Maevarano Formation was deposited [25].

The Anembalemba Member of the Maevarano Formation in the Berivotra Study Area ranges from 10 to 15 m in thickness. The member is primarily composed of two sandstone facies, Facies 1 and Facies 2. Facies 1 consists of light grey to white, fine- to coarse-grained, poorly sorted sandstone with small- to medium-scale tabular and trough cross-stratification and is thought to represent normal flow in an aggrading stream channel. Facies 2, on the other hand, consists of olive green, fine- to coarse-grained, clay-rich sandstone,

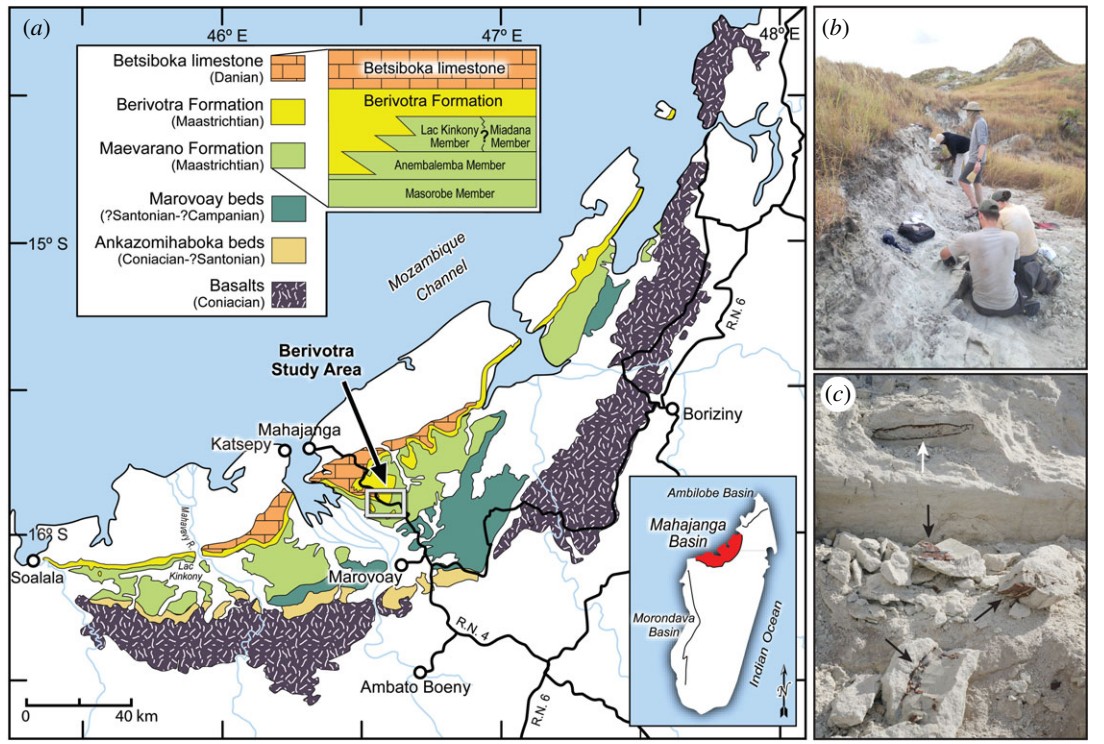

**Figure 1.** (*a*) Outcrop map of Late Cretaceous and Palaeocene strata in the Mahajanga Basin of northwestern Madagascar (see inset at lower right). The location of the Berivotra Study Area is highlighted by the rectangle. The holotype specimen of *Sahonachelys mailakavava* gen. et sp. nov. (UA 10581) was recovered from locality MAD05-38 in the Anembalemba Member of the Maevarano Formation (see stratigraphic relations in inset at upper left), which is of latest Cretaceous (Maastrichtian) age. (*b*) Locality MAD05-38, type locality of *Sahonachelys mailakavava*, view looking north-northwest. (*c*) Close-up view of quarry face at locality MAD05-38 showing longitudinal cross-section of UA 10581 above (white arrow, caudal end of specimen to the right) and large fragments of specimen in foreground (black arrows).

which has been interpreted as representing a debris flow setting [23,36]. To date, all articulated or associated vertebrate fossils, in addition to isolated specimens, in the Anembalemba Member have been recovered from Facies 2, while only isolated specimens have been recovered from Facies 1. UA 10581 was collected from a somewhat less massive variant of Facies 2 within locality MAD05-38. To date, the locality otherwise yielded over 100 fossil vertebrate specimens, including those of the crocodyliform *Miadanasuchus oblita*, the titanosaurian sauropod *Rapetosaurus krausei*, and the non-avian abelisauroid theropods *Majungasaurus crenatissimus* and *Masiakasaurus knopfleri*, in addition to possible paravian material.

# 3. Material and methods

## 3.1. Institutional abbreviation

UA, Université d'Antananarivo, Antananarivo, Madagascar.

## 3.2. Nomenclature

We here follow standard anatomical terminology for cranial structures of turtles [37], with modifications in regards to the carotid and facial nerve canal systems [38,39]. We also follow the recently outlined phylogenetic nomenclature for turtles [40]. All names are therefore phylogenetically defined clade names highlighted through the use of italics. For the undefined names *Araripemydidae* and *Podocnemidoidae*, we follow the taxonomic concepts of [41] and [3], respectively.

## 3.3. Micro-computed tomography scanning and three-dimensional modelling

High-resolution X-ray micro-computed tomography (μCT) scans were obtained for the cranium and hyoids of UA 10581, the holotype of *Sahonachelys mailakavava* gen. et sp. nov., at the University of

Texas High-Resolution X-ray Computed Tomography Facility using a North Star Imaging scanner with a 225 kV Feinfocus microfocal source. The specimen was scanned using a beam energy of 160 kV, a current of 0.2 mA, and an aluminium filter, resulting in an isometric voxel size of 0.0285 mm. The resulting slice images were segmented using the software Amira (http://www.amira.com) and three-dimensional models were exported in .ply format. The digital renderings used in the figures were compiled using the software Blender v. 2.71 (http://www.blender.org/). μCT-slice data as well as the three-dimensional models are deposited at MorphoSource (https://www.morphosource.org/Detail/MediaDetail/Show/media_id/88408).

## 3.4. Phylogenetic analysis

We explored the phylogenetic relationships of *Sahonachelys mailakavava* gen. et sp. nov. by scoring its morphology within the pleurodire matrix of [19], which, in turn, is based on previous work [3,5,6,42]. This matrix was chosen because it includes the most recent scoring for all known fossil pelomedusoids from the Late Cretaceous of Madagascar and India, in particular *Jainemys pisdurensis*, *Kinkonychelys rogersi*, *Kurmademys kallamedensis*, *Sankuchemys sethnai* and *Sokatra antitra*. The skull taxon *Caninemys tridentata* and the shell taxon *Stupendemys geographicus* were collapsed into a single terminal [17]. The matrix was expanded to include 21 recently developed characters based on μCT data [43]. Finally, a new character pertaining to the presence versus absence of an elongate posterior process of the maxilla was added to highlight similarities between *Sahonachelys mailakavava* and *Sokatra antitra*. The final matrix is provided in the electronic supplementary material and includes the full list of characters and character states.

The matrix was subjected to a parsimony analysis using the traditional search options of the software TNT [44]. Unless stated below, all default settings of the program were maintained. All characters that form morphoclines were ordered and are noted as such in the matrix, in particular characters 1, 10, 14, 18, 19, 51, 52, 56, 57, 75, 78, 81, 82, 86, 88, 95, 96, 99, 101, 103, 112, 114, 115, 119, 128, 130, 171, 172, 174, 175, 182, 183, 193, 195, 202, 220, 224, 225, 231 and 242 (counting from 1, not 0 per the format in TNT). *Proganochelys quenstedti* was selected as the outgroup. The memory was expanded to 10 000 trees. Light implied weighting was implemented with a $k$ value of 12 [45], and 1000 replicates of random addition sequences were followed by a round of tree bisection and reconnection.

## 3.5. Measure of relative neck length

We collected a dataset pertaining to the relative length of the neck of a sample of fossil and recent turtles. For this purpose, one of us (SWE) took measurement from pictures using ImageJ (https://imagej.nih.gov/ij/) for the cumulative length of all cervical centra and the midline length of the carapace. Relative neck length is herein expressed as the proportion of neck to carapace length in per cent. The length of missing neck vertebrae for fossil turtles was either estimated by reference to the length of preserved neighbouring vertebrae, excluding cervical I (the atlas), or by averaging and then extrapolating the length of preserved cervicals to the full column, again to the exclusion of cervical I. In either case, cervical I was disregarded, as it typically is much shorter than all subsequent vertebrae. If missing, the length of cervical I was estimated at one half of the length of cervical II. Given the numerous imprecisions associated with this methodology, all measurements and the resulting ratios should be seen as approximations that possibly deviate from the true measurement by a few percentage points. The final dataset for 68 specimens is provided in the electronic supplementary material.

# 4. Results

## 4.1. Systematic palaeontology

*Testudines* Batsch, 1788 [46]
*Pleurodira* Cope, 1865 [47]
*Pelomedusoides* Broin, 1988 [48]
*Sahonachelyidae* new clade name
**Registration number**. The clade name *Sahonachelyidae* is registered at RegNum with the number 570.
**Definition**. The largest extinct clade containing *Sahonachelys mailakavava* gen. et sp. nov.
**Reference phylogeny**. Figure 2.
**Diagnosis**. See species diagnosis below for characters pertaining to this clade.

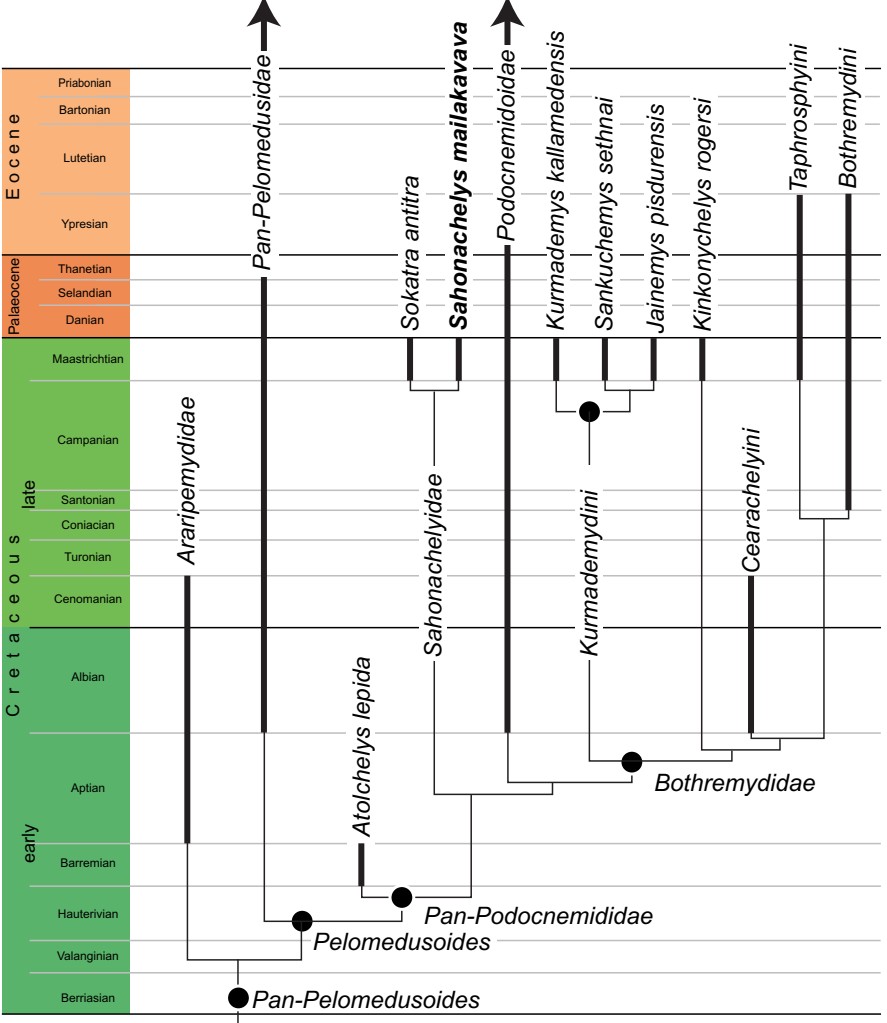

**Figure 2.** Time-calibrated strict consensus of 11 most parsimonious trees resulting from phylogenetic analysis. The full tree is provided in the electronic supplementary material.

*Composition*. *Sahonachelyidae* is currently hypothesized to consist of *Sahonachelys mailakavava* and *Sokatra antitra*, two extinct turtles from the Maastrichtian of Madagascar.

*Sahonachelys mailakavava* gen. et sp. nov.

*Type specimen*. UA 10581 (field number MAD 15359), a near complete skeleton consisting of the cranium (figures 3–6), mandible (figure 7), partial hyoids (figure 8), shell (figure 9), and remains of the limbs, girdles and vertebrae (figure 10).

*Type locality*. Locality MAD05-38, Berivotra Study Area, approximately 35 km southeast of the city of Mahajanga, Boeny Region, Madagascar. GPS coordinates are archived at the Denver Museum of Nature & Science and the University of Antananarivo and are available to qualified researchers.

*Type stratum.* Anembalemba Member, Maevarano Formation, dated to the Late Cretaceous (Maastrichtian, see §2 Geological setting).

*Nomenclatural acts*. This publication and its nomenclatural acts were registered at ZooBank on 16 December 2020, prior to publication. The LSID of the publication is urn:lsid:zoobank.org:pub:7E7E1F98-459F-41F6-A2BC-5CCF98DC3C3E, that of the new genus urn:lsid:zoobank.org:act:B5F0BA10-D35C-4739-A151-5AAD9D682193 and that of the new species urn:lsid:zoobank.org:act:813A5E08-6CDE-4FFA-8586-6AA2F8C069AA. The new clade name *Sahonachelyidae* was furthermore registered prior to publication at Regnum.

*Etymology*. The name *Sahonachelys mailakavava* combines the Malagasy words 'sahona' (pronounced sah-WHO-nah) for frog, 'mailaka' (pronounced my-LAH-kah) for quick, 'vava' (pronounced VAH-vah) for mouth, and the Greek word 'chelys' for turtle to mean 'quick-mouthed frog turtle' in allusion to the frog-like appearance of the skull and its inferred quick mode of suction feeding.

***Diagnosis***. *Sahonachelys mailakavava* gen. et sp. nov. can be diagnosed as a member of *Pan-Pelomedusoides* by presence of midline contact of prefrontals; presence of processus trochlearis pterygoidei; contact of postorbital with palatine resulting in formation of expanded septum orbito-temporale; involvement of prootic and/or quadrate in formation of foramen posterius canalis carotici interni; the absence of nasals, vomer and splenial; low-domed, oval shell; sutural contact of pelvis with shell; the absence of cervical scutes; reduction in neural count resulting in midline contact of posterior costals; broad plastron; equidimensional mesoplastra; single, median gular; and cervical column consisting of procoelous vertebrae. The shell of *S. mailakavava* can be distinguished from those of all other pan-pelomedusoids by the presence of elongate gular that broadly crosses entoplastron and fully hinders midline contact of extragulars and humerals (also present in the kurmademydine *Jainemys pisdurensis* and the taphrosphyines *Taphrosphys sulcatus* and *Ummulisani rutgersensis*) in addition to broad contribution of enlarged extragulars to margin of anterior plastral lobe resulting in reduced contribution from gular. The cranium of *S. mailakavava* can be distinguished from those of all members of the clade formed by *Podocnemidoidae* and *Bothremydidae* by plesiomorphic retention of deep temporal emarginations that results in reduced or absent squamosal-quadratojugal and parietal-quadratojugal contacts, retention of clear exposure of prootic in ventral view, laterally open foramen jugulare posterius, and the absence of cavum pterygoidei or fossa pterygoidea. The cranium of *S. mailakavava* can be distinguished from those of all remaining, more basal-branching representatives of *Pan-Pelomedusoides* (i.e. *Araripemydidae*, *Pelomedusidae* and *Atolchelys lepida*), but resembles those of *Sokatra antitra*, *Bothremydidae* and *Podocnemidoidae* in derived presence of posterior enclosure of incisura columella auris by quadrate to exclusion of Eustachian tube; flooring of processus paroccipitalis by expanded sheet of bone formed by basisphenoid, quadrate and basioccipital; quadrate-exoccipital contact; reduced ventral exposure of prootic (the latter two characters also present in *Araripemydidae*); short basioccipital; and exclusion of basioccipital from occipital condyle (absent in *Podocnemidoidae*). Among pan-pelomedusoids, *S. mailakavava* uniquely resembles *Sokatra antitra* by the presence of an elongate posterior process of the maxilla; formation of deeply interfingered sutural contact between jugal and maxilla; reduced ventral process of quadratojugal; formation of foramen posterius canalis carotici interni by basisphenoid, quadrate and prootic; and strong forward inclination of processus articularis. These characteristics serve to diagnose the clade Sahonachelyidae. *Sahonachelys mailakavava* differs from *Sokatra antitra* by having a much flatter and broader cranium with more dorsally oriented orbits, presence of prefrontal-palatine contact, absence of parietal-palatine contact, presence of narrower triturating surfaces that lack lingual ridge, shorter midline contact between palatines, presence of distinct supramaxillary artery sulcus on ventral side of jugal, and exposure of prootic anterior and posterior to foramen posterius canalis carotici interni.

## 4.2. Description

### 4.2.1. Cranium

The cranium of UA 10581 is exquisitely preserved in three dimensions (figure 3). Its midline length from the tip of the snout to the end of the occipital condyle is about 49 mm. The primary source of damage is taphonomic deformation and disarticulation to the left side of the cranium, resulting in a damaged temporal bridge and disarticulation between the quadrate and basicranium. The second source of damage is a freshly formed crack, created as a result of the specimen being discovered with heavy tools in the overburden at locality MAD05-38, that runs diagonally across the skull (figure 3*a*). Although the fracture was repaired during preparation, some fragments are missing along its margins, particularly near the posterolateral portion of the right pterygoid, the posterior tip of the right parietal, and the ventral portions of the left exoccipital. As all crushed, disarticulated, or missing portions are preserved on the opposite side of the cranium, we are able to assess the morphology of nearly all structures with confidence. A third, minor source of damage are irregularly placed pits and holes of varying sizes that are distributed across the ventral side of the palatines and pterygoids (figure 3*b*). As such marks are unknown in other turtles, we cautiously interpret them as taphonomic damage. Due to the fragility of the specimen, most of the matrix was not removed from the inside of the cranium, and the mandible and hyoids were left in place. The three-dimensional models generated from the μCT scan are therefore the most important source of anatomical information.

The cranium is notably flat anteriorly, particularly in comparison with its relatively short anteroposterior length and mediolateral breadth, but rises towards the posterior to give it a wedge shape in lateral view (figures 3*c* and 4*a,b*). The cranium is 49 mm long from the premaxillary labial

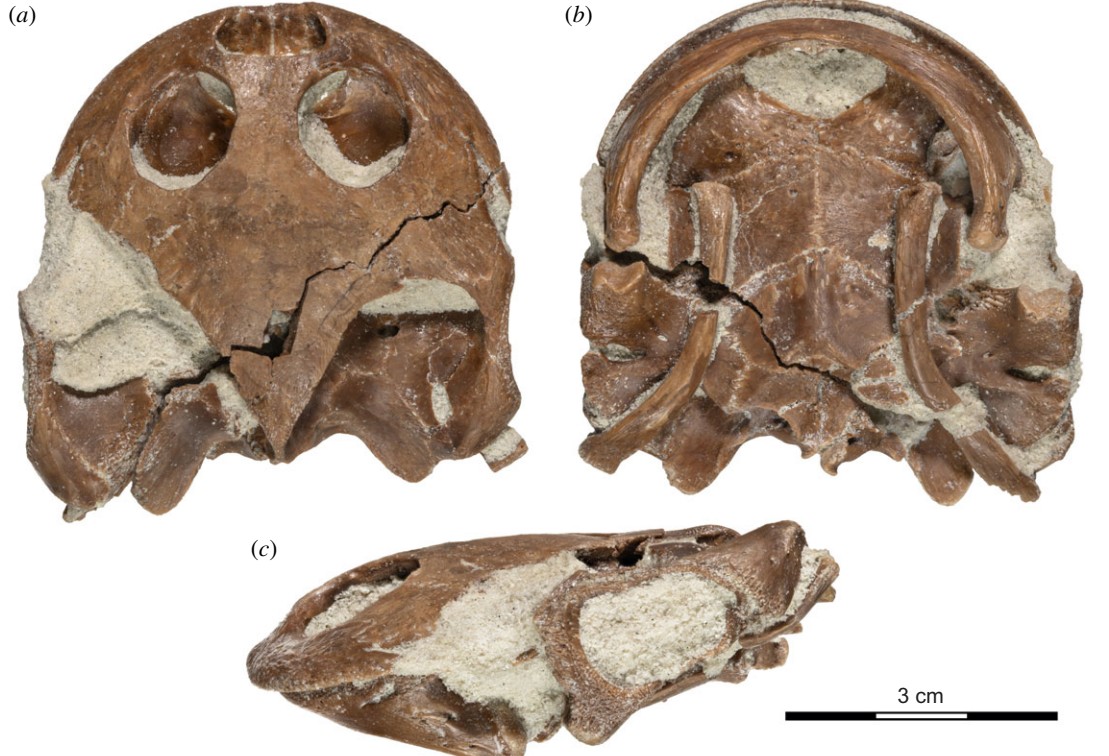

**Figure 3.** *Sahonachelys mailakavava* gen. et sp. nov., UA 10581, holotype, Late Cretaceous (Maastrichtian), Mahajanga Basin, Maevarano Formation, Madagascar. Photograph of skull with hyoid in (*a*) dorsal, (*b*) ventral and (*c*) left lateral views.

ridge to the basioccipital condyle, 16 mm high between the cranial roof and the posterior end of the basisphenoid, and 50 mm broad between the quadrates, measured just posterior to the incisura columella auris (all measurements taken digitally from the three-dimensional models and rounded to the nearest millimetre). The orbits are tightly spaced and are oriented dorsally more than they are either anteriorly or laterally. The lower temporal emargination rises to the level of the imaginary line that can be drawn between the lower margin of the orbit and the incisura columella auris. In contrast to most other turtles, the lower temporal emargination is partially framed anteroventrally by a well-developed posterior process of the maxilla. The upper temporal emargination is sufficiently deep to expose the anterior margin of the otic capsule.

*Nasal.* The external nares are preserved without any signs of damage (figures 3–5). There is no rugosity or suture along the external narial opening and we are therefore confident that nasals are absent in UA 10581.

*Prefrontal.* The prefrontal of UA 10581 is an elongate, rectangular element (figures 3–5). On the dorsal cranial surface, the prefrontal forms the dorsal margin of the low but broad external nares, anterolaterally contacts the maxilla, laterally forms the anteromedial quarter of the orbit, posteriorly contacts the frontal, and medially contacts its contralateral counterpart. The prefrontal forms a gracile descending process that contacts the palatine ventrolaterally along the orbital fossa. This contact is absent in *Sokatra antitra* [34]. In ventral view, a ventral ridge is apparent that visually separates the nasal capsule from the optic capsule. This ridge converges towards the posterior with its counterpart to form the sulcus olfactorius on the ventral side of the frontals.

*Frontal.* The frontal of UA 10581 is a small pentagonal element that is only slightly larger than the prefrontal (figures 3–5). On the dorsal cranial surface, the frontal anteriorly contacts the prefrontal along a transverse suture, laterally forms a broad contribution to the posteromedial portion of the orbit, posterolaterally contacts the postorbital along a parasagittal suture, posteriorly contacts the parietal along a transverse suture, and medially contacts its counterpart. Ventrally, the two frontals jointly form a distinct midline process that underlies the posterior two-thirds of the prefrontals. In addition, the frontal forms a distinct crista cranii, a ventral ridge that frames the anterior portion of the narrow sulcus olfactorius and merges posteriorly with the descending process of the parietal.

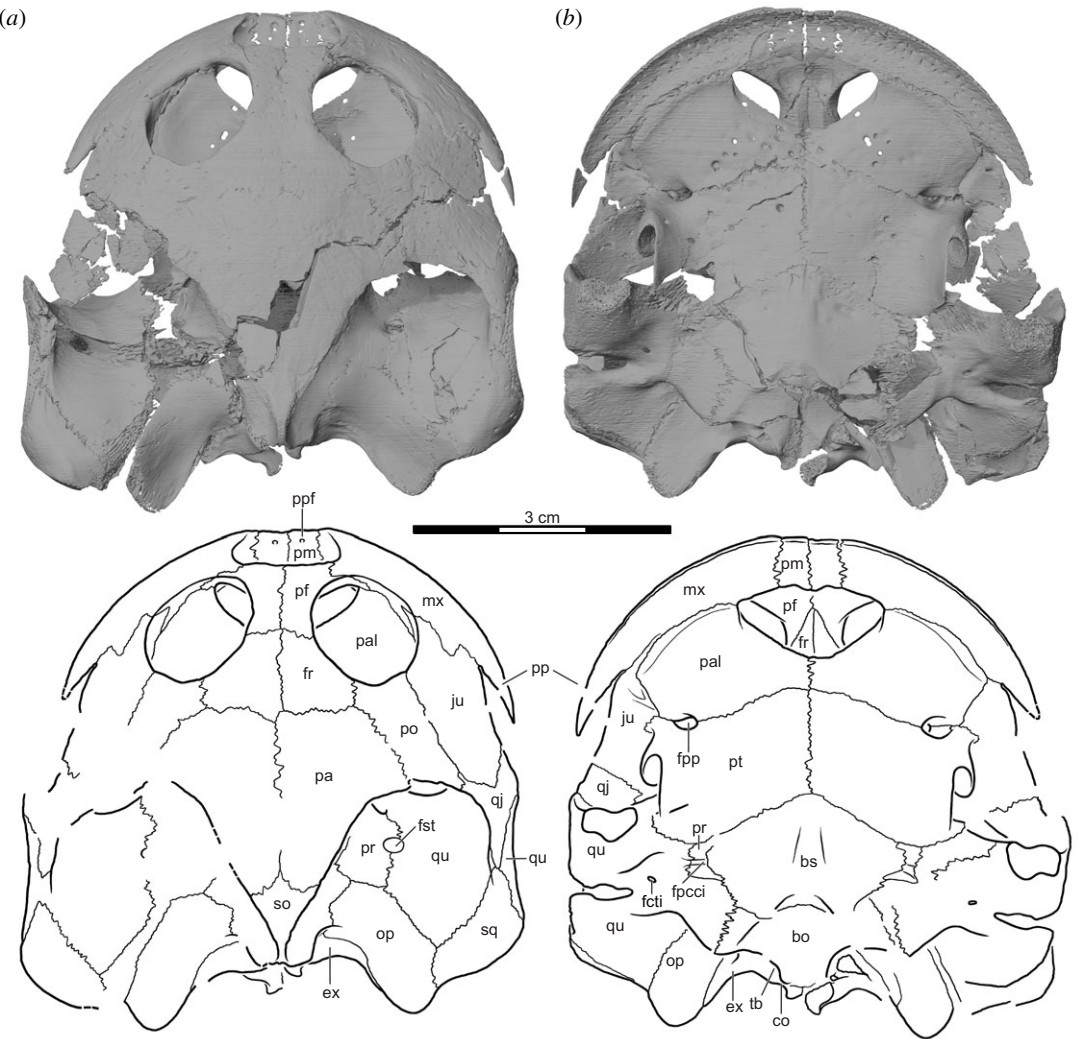

**Figure 4.** *Sahonachelys mailakavava* gen. et sp. nov., UA 10581, holotype, Late Cretaceous (Maastrichtian), Mahajanga Basin, Maevarano Formation, Madagascar. Three-dimensionally rendered model and line drawing of cranium in (*a*) dorsal and (*b*) ventral views. bo, basioccipital; bs, basisphenoid; co, condylus occipitalis; ex, exoccipital; fcti, foramen chorda tympani inferius; fpcci, foramen posterius canalis carotici interni; fpp, foramen palatinum posterius; fr, frontal; fst, foramen stapedio-temporale; ju, jugal; mx, maxilla; op, opisthotic; pa, parietal; pal, palatine; pf, prefrontal; pm, premaxilla; po, postorbital; pp, posterior process of the maxilla; ppf, praepalatine foramen; pr, prootic; pt, pterygoid; qj, quadratojugal; qu, quadrate; so, supraoccipital; sq, squamosal; tb, tuberculum basioccipitale.

*Parietal.* The parietal of UA 10581 is the largest bone on the dorsal surface of the cranium (figures 3–5). The parietal anteriorly contacts the frontal along a transverse suture, anterolaterally contacts the postorbital, posterolaterally frames the anteromedial portion of the upper temporal emargination, posteriorly contacts the supraoccipital, and medially contacts its counterpart. The dorsal plate of the parietal only slightly roofs the upper temporal fossa. Within the upper temporal fossa, the descending process of the parietal forms the anterodorsal margin of the trigeminal foramen, broadly contacts the prootic laterally, and contacts the supraoccipital posteriorly. Anterior to the trigeminal foramen, the descending process of the parietal forms the extensive anterior extension of the lateral braincase wall. This extension is approximately as long as the postorbital bar. The descending process is expanded at its base, but only contacts the pterygoid ventrally. A contact with the palatine is clearly absent on both sides of the cranium, in contrast to the condition in *Sokatra antitra* [34]. The posterior margin of the foramen interorbitale formed by the descending branch of the parietal is dorsally confluent with the crista cranii of the frontals, which form the sulcus olfactorius.

*Jugal.* The jugal of UA 10581 is a relatively large element that forms much of the cheek region of the cranium (figures 3–5). Due to the flattened nature of the cranium, its external surface is mostly oriented

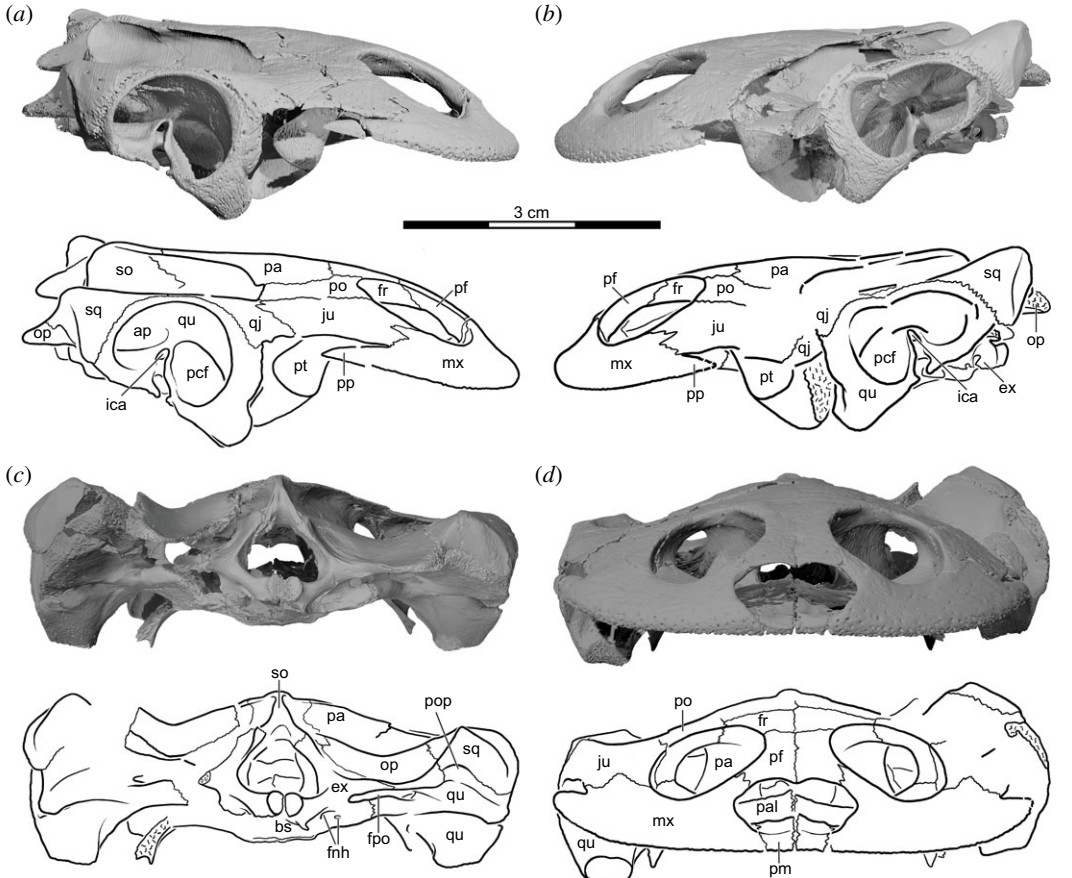

3 cm

**Figure 5.** *Sahonachelys mailakavava* gen. et sp. nov., UA 10581, holotype, Late Cretaceous (Maastrichtian), Mahajanga Basin, Maevarano Formation, Madagascar. Three-dimensionally rendered model and line drawing of skull in (*a*) right lateral, (*b*) left lateral, (*c*) posterior and (*d*) anterior views. ap, antrum postoticum; bs, basisphenoid; ex, exoccipital; fnh, foramen nervi hypoglossi; fpo, foramen postoticum; fr, frontal; ica, incisura columella auris; ju, jugal; mx, maxilla; op, opisthotic; pa, parietal; pal, palatine; pcf, precolumellar fossa; pf, prefrontal; pm, premaxilla; po, postorbital; pop, posterior opening of antrum postoticum; pp, posterior process of the maxilla; pt, pterygoid; qj, quadratojugal; qu, quadrate; so, supraoccipital; sq, squamosal.

dorsally, not laterally. On the dorsal side of the cranium, the external plate of the jugal broadly contacts the maxilla anteroventrolaterally, forms the anterior half of the lower temporal emargination, posteriorly contacts the quadratojugal, posteromedially contacts the postorbital along an elongate suture, and anteromedially forms the posterolateral sixth of the margin of the orbit. The jugal-maxilla contact is peculiar in that the suture is dominated by a single, large posterior spike-like process of the maxilla that is received by a respective embayment of the anterior jugal margin, an unusual feature also found in *Sokatra antitra* [34]. A jugal contribution to the upper temporal emargination is precluded by a clear, though short contact of the postorbital with the quadratojugal. A jugal-quadrate contact is also absent. Within the orbit, the short medial process of the jugal inserts anteriorly into the maxilla, broadly contacts the palatine medially, and has a short posterior contact with the postorbital. Furthermore, the medial process of the jugal contacts the anterodorsally upturned trochlear process of the pterygoid. A small opening, interpreted as the entry foramen for the supramaxillary artery, is framed by the jugal laterally, the palatine medially and ventrally, the pterygoid posteriorly, and the postorbital dorsally (electronic supplementary material, figure S1a). The anterior pathway of the supramaxillary artery can be followed along a deep groove that is incised into the ventral surface of the jugal. This sulcus would have been broadly exposed ventrally even if the palatine were more closely articulated to the jugal than preserved. The supramaxillary artery sulcus bifurcates, best observed on the ventral side of the right jugal in the three-dimensional model (electronic supplementary material, figure S1a). This is more clearly developed on the right element. A short subordinate branch fades laterally on the ventral surface of the jugal. The second, larger subordinate branch curves anteriorly and enters the maxilla at the triple junction of maxilla, jugal and palatine via

a relatively large foramen. This structural arrangement for the entry of the supramaxillary artery is unusual among turtles [37]. The jugal of UA 10581 does not contribute to the triturating surfaces.

*Quadratojugal.* The quadratojugal of UA 10581 is a small element that has a broad anterolateral contact with the jugal, a short anterodorsomedial contact with the postorbital, and a broad posterior contact with the quadrate (figures 3–5). A short posterior contact with the squamosal is clearly preserved on the right side of the cranium, but was probably absent on the left, where this area is also well preserved. This quadratojugal-squamosal contact is absent in *Sokatra antitra* [34]. The quadratojugal of UA 10581 otherwise contributes to the upper and lower temporal emarginations. The quadratojugal does not contact the maxilla due to the broad contribution of the jugal to the lower temporal emargination. The quadratojugal also does not contact the parietal due to the postorbital extending posteriorly to contribute to the upper temporal emargination. Ventrally, the quadratojugal extends only halfway down the height of the cavum tympani of the quadrate. In many turtles, the quadratojugal extends farther ventrally to form a much larger contact with the quadrate than that in UA 10581 [37]. There is a roughened area along the anterior margin of the quadrate that superficially resembles an articulation site for the quadratojugal, potentially suggesting the presence of a ventral process of the quadratojugal that extends along this portion of the quadrate. However, the near-perfect preservation of the quadratojugal on the right side suggests that the quadratojugal indeed lacks a ventral process and that the quadratojugal-quadrate contact is not as extensive as that in other turtles. We interpret the roughened area on the quadrate as the insertion site for the quadratomaxillary ligament, broadly found in crown turtles [49]. A similar condition is also found in *Sokatra antitra* [34].

*Squamosal.* The squamosal of UA 10581 is a patelliform element that caps the posterolateral portions of the greatly enlarged antrum postoticum (figures 3–5). It broadly contacts the quadrate anteromedially and ventrally and briefly contacts the opisthotic posteromedially. A short anterior contact with the quadratojugal above the cavum tympani is developed on the right side of the cranium, but not on the left side. This contact is absent in *Sokatra antitra*. A peculiar, slit-like opening along the triple-junction of the squamosal, quadrate and opisthotic is present at the anterolateral tip of the paroccipital process, which creates an opening from which the antrum postoticum can be seen in ventrolateral view (electronic supplementary material, figure S1b). It is unclear if this opening is also present in *Sokatra antitra*. The preservation of the cranium in this area is pristine and we thus interpret this opening to be a genuine feature. We are not aware of any other turtle that has a comparative opening into the antrum postoticum. On the posterodorsal tip of the squamosal, three ridges converge upon a blunt posterior process: a faint, anteroposteriorly directed dorsal ridge that separates the upper temporal fossa from the outside of the cranium, a ventrolaterally directed lateral ridge that outlines the anterodorsal margin of the depressor mandibulae, and a roughly transversely oriented ventral ridge that is confluent with the paroccipital process (electronic supplementary material, figure S1c). On the lateral surface of the cranium, the squamosal broadly contacts the quadrate anteroventrally, but does not contribute to the margin of the cavum tympani.

*Postorbital.* The postorbital of UA 10581 is an elongate bone that consists of a dorsal plate and a short ventral process (figures 3–5). On the cranial surface, the postorbital forms the posterior margin of the orbit, broadly contacts the jugal laterally, has a short posterolateral contact with the quadratojugal, broadly contributes to the upper temporal emargination, and broadly contacts the parietal posteromedially and the frontal anteromedially. The ventral process of the postorbital forms the narrow posterolateral wall of the orbit, the septum orbito-temporale. Within the orbit, the postorbital contacts the frontal dorsomedially, the jugal anteriorly and the palatine ventrally. The lateral process of the pterygoid rises posterior to the foramen palatinum posterius to contact the dorsal plate of the postorbital from below.

*Premaxilla.* The premaxilla of UA 10581 is a paired, flat element that forms the anterior-most margin of the cranium (figures 3–5). It contacts the maxilla laterally and its contralateral counterpart medially. A posterior contact with the vomer is absent, as the latter is not developed. The premaxilla forms the ventral floor of the nasal capsule and the indistinct ventral margin of the confluent external nares. A number of small foramina pierce the premaxilla. We interpret those foramina, which are associated with a posteriorly oriented dorsal groove, as the praepalatine foramina. Ventrally, the premaxilla forms the median aspects of the low labial ridge and forms the anterior margin of the confluent internal nares. The premaxillae jointly form a posteriorly oriented midline process that rises above the triturating surface and probably contacted the cartilaginous septum that separated the internal nares along the midline.

*Maxilla.* In contrast to most pelomedusoids, the maxilla of UA 10581 is a notably gracile element (figures 3–5). The maxilla forms the anterolateral margin of the cranium, anteromedially contacts the premaxilla, anterodorsomedially contacts the prefrontal above the external nares and anterior to the orbit, and posterodorsomedially contacts the jugal along a suture that is dominated by a single,

posteriorly oriented maxillary process (see *Jugal* above). The maxilla forms the lateral margin of the external nares, as well as the anterolateral margin of the orbit. Within the nasal cavity, each maxilla has two well-developed, unnamed foramina that are connected to form a short canal. Internally, this short canal connects with the canal for the supramaxillary artery, which traverses the maxilla anteroposteriorly and has its posterior foramen at the maxilla-jugal-palatine contact (see *Jugal*). The two foramina within the nasal cavity are located immediately posterior to the maxillary margin of the external nares (anterior foramen) and just anterior to the maxillary contact with the palatine (posterior foramen). We are not aware of any other pelomedusoids that possess this medial canal along the inner surface of the nasal cavity, but this may be due to a lack of comparative documentation within the group. The orbito-narial bar is gracile relative to the size of the orbit. Within the orbit, the maxilla broadly contacts the palatine medially, but only forms the most lateral aspects of the orbit floor. A contribution to the medially open foramen orbito-nasale is precluded by a contact of the prefrontal with the palatine within the orbit. A notable feature of the maxilla is its posterior process, a spine-like outgrowth that extends lateral and ventral to the ventral margin of the jugal, but does not contact the jugal. The posterior process of the maxilla therefore projects into the cheek emargination. Although the presence of such a process is hinted at in some testudinids, such a pronounced posterior process is otherwise only developed in *Sokatra antitra* [34]. This structure possibly represents the partial ossification of the quadratomaxillary ligament found in extant turtles [49]. In ventral view, the maxilla is conspicuous by being mostly oriented mediolaterally, not anterolaterally, as in the vast majority of turtles. In this view, the maxilla is sickle-shaped and has a short anteromedial contact with the premaxilla, an elongate posteromedial contact with the palatine, and a short posterior contact with the jugal. Contact with the vomer is absent, as this element is not developed. The maxilla, along with the premaxilla, forms a low labial ridge that evenly extends along the entire margin of the palate, including the posterior process. The ridge lacks teeth, notches, or sinuosities. The triturating surface is mostly formed by the maxilla, but includes minor contributions from the premaxilla and palatine (see below). The triturating surface is flat, is marked by nutritive foramina, and, when the posterior process is disregarded, retains equal width along its entire length.

*Vomer.* The exceptional preservation of the specimen strongly suggests that the vomer is not developed in UA 10581. The presence of cartilaginous remnants that subdivide the internal nares are possibly hinted at by the joint posterior process formed by the premaxillae. Although initially reported as absent [34], it is possible that a remnant of the vomer is present in *Sokatra antitra* (see Discussion below).

*Palatine.* The palatine of UA 10581 is an extremely thin bone that forms much of the primary palate (figures 3–5). The bone is punctured by a number of peculiar pits and holes that are not found in other turtles and that we believe to be taphonomic damage (see §4.2.1). In ventral view, the palatine broadly contributes to the posterolateral margin of the internal choanae anteromedially, broadly contacts the maxilla anterolaterally, contacts the jugal posterolaterally, broadly contacts the pterygoid posteriorly, and contacts its counterpart medially. The palatine narrowly contributes to the triturating surface along its full contact with the maxilla. This contribution increases in width towards the posterior, but remains minor even posteriorly. A groove that we interpret as holding the supramaxillary artery runs parallel to the contact with the jugal (see *Jugal* above). The lenticularly shaped foramen palatinum posterius is located within the palatine-pterygoid contact. The palatines are moderately arched dorsally along their midline contact to roof the narial passage. In dorsal view, the palatine forms much of the orbital floor, anteromedially contributes to the enlarged and confluent foramen orbito-nasale, narrowly contacts its counterpart along the midline, and contacts the pterygoid posteromedially, the descending process of the postorbital laterodorsally, the jugal laterally and the maxilla anterolaterally. A dorsal contact with the parietal is clearly absent. A low ridge along the dorsal extension of the septum orbito-temporale delimits the posteroventral margin of the orbit. The foramen palatinum posterius is situated posterior to this ridge.

*Quadrate.* The quadrate of UA 10581 is a complex bone that contributes to a series of structures (figures 3–5). Within the upper temporal fossa, the quadrate contacts the prootic anteromedially, the opisthotic posteromedially and the squamosal posterolaterally. The quadrate forms the lateral half of the short, but relatively large canalis stapedio-temporalis. Although the trigeminal region is slightly damaged and disarticulated on both sides of the cranium, it is clear that the quadrate, together with the prootic, pterygoid and parietal formed the trigeminal foramen. Specifically, the quadrate and prootic form a sheet-like lamina that projects over the large cavum epiptericum to form the posterior and lateral margins of the trigeminal foramen. The space of the cavum is created, in part, by the mediolaterally broad anterior surface of the prootic. The trigeminal nerve branches would have

traversed this fossa mediolaterally, whereas the vena capitis lateralis enters the cavum from the posterior through the foramen cavernosum, which is dorsally overhung by the sheeted lamina formed by the prootic and quadrate (electronic supplementary material, figure S1d). We are unaware of any other turtle where the quadrate contributes to the trigeminal foramen [37]. In ventral view, the quadrate forms the reniform mandibular condyle, anteromedially contacts the pterygoid, medially contacts the prootic and basisphenoid, broadly contacts the basioccipital ventral to the fenestra postotica and the exoccipital along the dorsal margin of the fenestra postotica, as well as the opisthotic dorsal to the fenestra postotica. Foramina chorda tympani inferius for the chorda tympani branch of the facial nerve are developed posteromedial to the base of the mandibular condyle. The mandibular condyle facets are strongly sloped anteroventrally, rather than strictly ventrally. As a result, the mandibular articulation is located below the anterior margin of the cavum tympani. The articulation with the mandible is described in detail below (see *Articular*). The flat medial process of the quadrate broadly floors the cavum acustico-jugulare. A minor depression medial to the mandibular condyle leads to the canal that leads to the foramen posterius canalis carotici interni, which is located largely within the prootic but is laterally framed by the quadrate and medially by the basisphenoid. A medial contact with the exoccipital is furthermore apparent in posterior view just below the fenestra postotica. In lateral view, the quadrate forms the voluminous cavum tympani. In this view, the quadrate anterodorsally contacts the quadratojugal and posterodorsally contacts the squamosal. A dorsal contribution to the upper temporal emargination is only developed on the left side of the cranium (see *Squamosal* or *Quadratojugal*). This contribution is present on both sides of the cranium in *Sokatra antitra* [34]. The quadrate otherwise broadly contributes to the posterior margin of the lower temporal emargination. The margins of the ovoid cavum tympani are formed by the quadrate, with the exception of a short, posteroventral gap. The cavum tympani itself possesses two voluminous pouches, a notably broad and deep precolumellar fossa formed entirely by the quadrate and the antrum postoticum, which is formed jointly with the squamosal. Along its medial half, the quadrate forms a posteriorly enclosed incisura columella auris. Along its lateral half, the quadrate otherwise forms a laterally open notch, which would have allowed for the passage of the Eustachian tube from the pharynx to the middle ear. The Eustachian tube was interpreted as being separated from the incisura columella auris by a bony wall in *Sokatra antitra* [34], but we here interpret that as an error (see Discussion below). Ridges that run from the lateral edge of the incisura along this notch define the above-mentioned pouches. A very unusual feature of the quadrate of UA 10581 is its contribution to the lateral enclosure of the lateral semicircular canal (LSC) of the endosseous labyrinth, which is best seen when three-dimensional models of the prootic and opisthotic are seen in lateral view (electronic supplementary material, figure S1e). The LSC of turtles is usually formed exclusively by the prootic, supraoccipital and opisthotic, and a quadrate contribution has, to our knowledge, not been observed for any other turtle [37], including those documented by a large set of µCT scans that were specifically collected to investigate the inner ear morphology of turtles (largely unpublished data, but see, for instance, [50]).

*Pterygoid.* The pterygoid of UA 10581 is a large element, the ventral surface of which is marked by unusual round pits that we interpret as taphonomic damage (see §4.2.1 above). In ventral view, the pterygoids jointly form an anteriorly oriented chevron, which protrudes anteriorly between the palatine (figures 3–6). The pterygoid anteriorly contacts the palatine both medial and lateral to the foramina palatinum posterius, anterolaterally contacts the jugal, laterally forms the processus trochlearis pterygoidei, posterolaterally contacts the quadrate, posteriorly contacts the prootic, posteromedially contacts the basisphenoid, and medially contacts its counterpart. In ventral view, the surfaces of the pterygoid trochlea are mostly oriented anteroposteriorly and are therefore oriented parallel to another, instead of forming an angle. The anterior end of the trochlear process is strongly tilted dorsally such that the structure has a nearly vertical orientation that is unparalleled in other pleurodires. The external process of the pterygoid reaches dorsally to briefly contact the palatines lateral to the foramen palatinum posterius, laterally the jugal, and posteriorly the postorbital. Posteriorly, the trochlea is confluent with a downwardly curved pterygoid flange that occupies the posterolateral third of the pterygoid. This flange would reach the level of the mandibular condyle if the specimen were still fully articulated in this region. Neither a cavum pterygoidei or fossa pterygoidea are developed. On its dorsal surface, the pterygoid broadly contacts the descending process of the parietal along the elevated crista pterygoidea. The crista is dorsally highest near the trigeminal foramen, where it comes close to contacting the anteromedial process of the prootic. The crista pterygoidei is not strictly oriented sagittally but extends laterally towards the quadrate with its posterior end. Together with the quadrate and prootic, the crista pterygoidei hereby forms a large cavum epiptericum, which is the extracranial

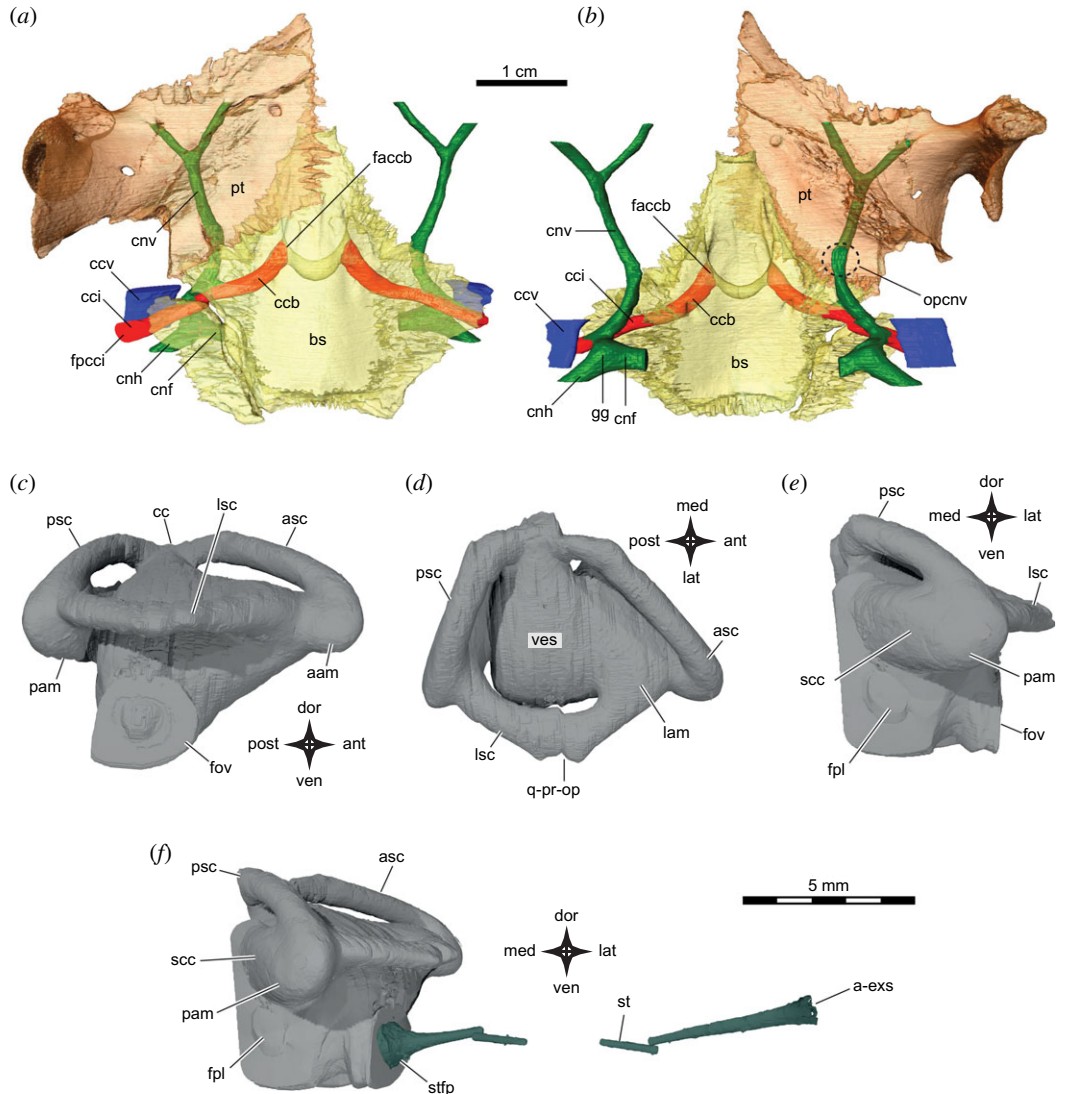

**Figure 6.** *Sahonachelys mailakavava* gen. et sp. nov., UA 10581, holotype, Late Cretaceous (Maastrichtian), Mahajanga Basin, Maevarano Formation, Madagascar. Three-dimensionally rendered models of the carotid and facial nerve canal system in (*a*) ventral and (*b*) dorsal views, of the right inner ear in (*c*) right lateral, (*d*) dorsal and (*e*) posterior views, and of the right inner ear and right stapes in (*f*) posterior view. a-exs, articulation site of extrastapes; aam, anterior ampulla; asc, anterior semicircular canal; bs, basisphenoid; cc, common crus; ccb, canalis caroticus basisphenoidale; ccf, canalis caroticus facialis; cci, canalis caroticus internus; ccv, canalis cavernosus; cnh, canalis nervus hyomandibularis; cnv, canalis nervi vidiani; faccb, foramen anterius canalis carotici basisphenoidalis; fov, fenestra ovalis; fpcci, foramen posterius canalis caroticі interni; fpl, fenestra perilymphatica; gg, geniculate ganglion; lam, lateral ampulla; lsc, lateral semicircular canal; opcnv, opening in the roof of the canalis nervi vidiani; pam, posterior ampulla; psc, posterior semicircular canal; pt, pterygoid; q-pr-op, quadrate-prootic-opisthotic junction highlighting small quadrate contribution to lateral semicircular canal; scc, secondary common crus; st, stapes; stfp, stapedial footplate; ves, vestibule.

space in which the trigeminal ganglion is located and into which the foramen cavernosum drains the vena capitis lateralis coming from the canalis cavernosus. The trigeminal area is disarticulated on both sides of the specimen, but a rounded notch at the posterior limit of the crista suggests that the pterygoid forms the anteroventral margin of the trigeminal foramen. The pterygoid forms a broad floor that is medially directed towards the basisphenoid. This floor, although not distinctly recessed to form a groove, serves as the sulcus cavernosus along which the vena capitis lateralis traverses the cranium. A canal extends anteroposteriorly through the pterygoid below the floor of the sulcus cavernosus. The canal pierces the prootic posteriorly to connect with the internal carotid canal. Anteriorly the canal splits into two canals that exit just behind the orbit and the foramen palatinum posterius, and anterolateral to the base of the crista pterygoidei. The canal rises from below to also broadly puncture the flooring of the sulcus

cavernosus, best seen in the three-dimensional models (figure 6*b*). The resulting opening is formed on both sides of the cranium by the pterygoid and prootic and has an elongate, oval outline. This canal system is clearly identifiable as pertaining to the vidian nerve, as it connects posteriorly to the internal carotid canal, which itself is connected to the facial nerve canal in the prootic (figure 6*a*,*b*). The function of the aforementioned foramen in the floor of the sulcus cavernosus is unclear to us, however, as no branch of the vidian nerve is known to exit in this position in any turtle [39]. Although extant pleurodires lack the palatine artery [39,51], contra [43], the respective foramen of UA 10581 is topologically located in the position where a foramen for the palatine artery would be expected, namely the floor of the sulcus cavernosus. In turtles that have a palatine artery, the respective foramen anterius canalis carotici palatinum is usually formed between the basisphenoid and pterygoid, making the positions of the possible palatine artery foramen somewhat unusual. However, in those turtles with a palatine artery, the basisphenoid and pterygoid bones also form a narrow sulcus cavernosus that differs strongly from the broad sulcus cavernosus morphology seen in UA 10581. Thus, the topological difference of this foramen relative to the palatine foramina of other turtles could possibly be explained by differences in the structural architecture of UA 10581. Although we cannot know for certain if the palatine artery was present in UA 10581, we herein label this canal as the vidian nerve canal based solely in reference to the fact that extant outgroups lack this artery [39,51].

*Supraoccipital.* As in all pleurodires, the supraoccipital of UA 10581 makes a small contribution to the posterior limit of the dorsal cranial roof where it contacts the parietals anteriorly (figures 3–5). Within the upper temporal fossa, the supraoccipital contacts the parietals anteriorly, the prootics anterolaterally, the opisthotics laterally and the exoccipitals posterolaterally. Internally, the supraoccipital forms the common crus portion of the endosseous labyrinth. A true supraoccipital crest is absent, as the supraoccipital does not extend posteriorly beyond the level of the foramen magnum. At its posterior end, the supraoccipital forms two ridges that connect ventrally with the exoccipitals. This part of the supraoccipital also forms the dorsal roof of the foramen magnum.

*Exoccipital.* The exoccipitals of UA 10581 frame the lateral and ventral aspects of the foramen magnum (figures 3–5). The occipital condyle is entirely formed by the exoccipitals, but only protrudes slightly beyond the tubercula basioccipitale, which are formed by the basioccipital. Dorsally, the exoccipital contacts the supraoccipital. Laterally, the exoccipital forms a posteriorly inclined process that abuts against the posterior surface of the paroccipital process of the opisthotic. This process is connected with the dorsal process that contacts the supraoccipital via a ridge, which starts shallowly laterally and becomes more pronounced on the dorsal process. Here, the ridge has a relatively acute margin and is dorsally confluent with the bifurcated ridge of the supraoccipital. The posterolateral process of the exoccipital forms portions of the dorsal margin of the fenestra postotica, which is undivided (i.e. coalescent with the foramen jugulare posterius), and dorsoventrally narrow and thus slightly slit-like. The exoccipital also extends laterally in the floor of the cavum acustico-jugulare and contacts the quadrate laterally. Within the cavum acustico-jugulare, the exoccipital forms the back wall of the recessus scalae tympani and the posterior margin of the foramen jugulare anterius, both of which are extremely wide and voluminous. Along its entire ventral surface, the exoccipital rests on top of the basioccipital. Posterolaterally, a pair of hypoglossal foramina are developed along the suture with the basioccipital on either side. These foramina are hidden from ventral view by a moderate overhang formed by the basioccipital. The posterior foramen is at least twice as wide as the more anteriorly positioned foramen, suggesting that three rami of the hypoglossal nerve were once present and the posteriormost two nerves shared a single aperture.

*Basioccipital.* The basioccipital of UA 10581 is partially damaged, but all contacts and structures can confidently be assessed (figures 3 and 4). In ventral view, the basioccipital contacts the basisphenoid anteriorly along a slightly sinuous transverse suture, the quadrates anterolaterally and the exoccipitals posterodorsolaterally. Posterolaterally, this element roofs the hypoglossal foramina. The hypoglossal foramina are situated at the basioccipital-exoccipital suture, but are mostly situated within the exoccipitals. The basioccipital is about twice as broad as it is long. The basioccipital does not contribute to the occipital condyle, which is rare among turtles, but common in pelomedusids [37] and bothremydids [5]. The basioccipital of UA 10581 forms a pair of indistinct tubercles that frame a shallow median depression. The basioccipital furthermore forms a broad, semi-lunate depression that extends anteriorly onto the basisphenoid.

*Prootic.* The prootic of UA 10581 forms the anterolateral part of the braincase and is broadly exposed within the upper temporal fossa, where it contacts the quadrate laterally, the opisthotic posteriorly, the supraoccipital posteromedially and the parietal anteromedially (figures 3–5). The medial half of the short but large stapedio-temporal canal and foramen are formed by the prootic in the anterior portion

of the otic capsule. Internally, the canalis stapedio-temporalis leads into the cavum acustico-jugulare, which has peculiar subdivisions formed by delicate laminae of the prootic for the various nerves and vessels that traverse this structure. For instance, the course of the vena capitis lateralis is clearly marked by a horizontal lamina that floors its course even before the quadrate frames this vein laterally to completely encase the canalis cavernosus. The flooring lamina for the vena capitis lateralis suggests that the vein entered the dorsal part of the cavum acustico-jugulare, because the lamina is positioned dorsally to the level of the fenestra ovalis. This is consistent with dissection data from pleurodires [37], which indicates that the vein in *Podocnemis* spp. is closely associated with the underside of the opisthotic, and thus the dorsal part of the cavum. Anteriorly, the canalis cavernosus extends between the prootic and quadrate, and exits into a broad fossa, the extracranial cavum epiptericum, that is formed between these two bones, the pterygoid and, to a lesser extent, the parietal. The cavum epiptericum is hypertrophied by the broad extent of the basal process of the prootic. This basal process is medially pierced by a single, large acoustic foramen that extends into the cavum labyrinthicum, as well as by the medial foramen for the canalis nervus facialis. All of these medial foramina are positioned within the relatively deeply recessed fossa acustico-facialis. The canalis nervus facialis traverses the prootic laterally from this fossa, until it bifurcates, indicating that the position of the geniculate ganglion lies within the prootic. The geniculate ganglion is positioned below the cavum labyrinthicum but above the carotid canal system. A short vertical canal directs the vidian nerve from the ganglion position into the canalis caroticus internus. A second canal, the canalis nervus hyomandibularis distalis for the hyomandibular branch of the facial nerve, is laterally continuous with the mediolateral trajectory of the canalis nervus facialis, and exits within the cavum acustico-jugulare below the level of the vena capitis lateralis. The exit foramen of the canalis nervus hyomandibularis distalis is framed by distinct ridges of the prootic, highlighting the subdivision of the cavum acustico-jugulare. A vertical ridge that forms the posterior wall of the foramen for the canalis nervus hyomandibularis distalis simultaneously forms a broad posterior surface immediately lateral to the prootic portion of the fenestra ovalis. This surface is distinctly recessed, and the respective fossa has been described for a number of turtles under different terminologies [52,53]. We use the name perilymphatic fossa [54] given that the fossa is probably associated with the perilymphatic system of the turtle inner ear. The fenestra ovalis itself is only incompletely surrounded by bone, as the prootic and opisthotic have no contact in the ventral margin of the foramen. Instead, the fenestra ovalis is ventrally open towards the quadrate, which floors this part of the cavum acustico-jugulare but does not have direct contact with the processus interfenestralis of the opisthotic. A noteworthy observation of the prootic part of the cavum labyrinthicum is that the prootic forms a portion of the elongate LSCs, a feature that is probably a synapomorphy of pleurodires, as it is seldom seen in cryptodires [52]. The prootic is exposed and integrated into the ventral cranial surface between the quadrate laterally and posteriorly, the pterygoid anteriorly and the basisphenoid medially. The ventral side of the prootic forms a sulcus for the internal carotid artery that extends laterally onto the quadrate. The sulcus continues medially into the cranium, but is floored by the basisphenoid to form a true canal. Thus, the foramen posterius canalis caroticus internus is formed by the quadrate, prootic and basisphenoid. Almost immediately after being completely covered by bone, the canalis nervus vidianus enters the canalis caroticus internus from above. The vidian nerve then extends through the same canal as the internal carotid artery for a short distance. Between the prootic and basisphenoid, the canalis internus caroticus bifurcates into a medial branch that enters the basisphenoid and that can confidently be identified as the canalis caroticus cerebralis and a second, more lateral branch that is anteriorly directed and carries the vidian nerve and possibly also the palatine artery (see *Pterygoid* above). Although it is possible that this canal represents the canalis caroticus palatinum, we here interpret it as the canalis nervus vidianus by reference to the extant outgroups [39,51].

*Opisthotic.* The opisthotic of UA 10581 forms parts of the otic capsule and braincase medially and laterally forms a wing-like, posterolaterally directed paroccipital process that is dorsoventrally flat (figures 3–5). The dorsal surface of this process is broadly exposed in the floor of the upper temporal fossa, where it contacts the prootic anteriorly, the quadrate anterolaterally, the squamosal laterally, the supraoccipital medially and the exoccipital posteromedially. The tip of the paroccipital process forms a rounded extension that protrudes posteriorly far beyond the level of the occipital condyle and squamosal. The opisthotic forms the dorsal margin of the fenestra postotica, which is otherwise formed by the quadrate laterally and the exoccipital medially. The medial side of the opisthotic is dorsoventrally expanded with regard to the paroccipital process, and forms the posterior portion of the cavum labyrinthicum. The posterior third of the fenestra ovalis is framed by the processus interfenestralis, which descends almost vertically from the remainder of the opisthotic. The processus

interfenestralis is notably slender, as in many pleurodires, but does not contact the floor of the basicranium ventrally, which is formed by the basioccipital and quadrate in this part of the cranium. Thus, a narrow hiatus postlagenum remains between the opisthotic and braincase floor. The fenestra perilymphatica is notched into the medial margin of the processus interfenestralis, but remains unossified medially and is thus open toward the cavum cranii proper. The recessus scalae tympani and foramen jugulare anterius, which are formed between the opisthotic anteriorly and the exoccipital posteriorly, are notably large in diameter. The glossopharyngeal nerve usually traverses the opisthotic near the base of the processus interfenestralis, but this foramen is not evident to us in µCT scans of UA 10581. As the processus interfenestralis of UA 10581 is particularly thin and slender, and the foramen jugulare anterius is particularly broad, it seems possible that the glossopharyngeal nerve exited through the foramen jugulare anterius, which is plausible as this is also the common course for this nerve in non-chelonian reptiles [55].

*Basisphenoid.* The basisphenoid of UA 10581 broadly contacts the pterygoids anteriorly, the prootics laterally, the quadrates posterolaterally and the basioccipital posteriorly (figures 3–6). The ventral surface of the bone gently slopes upwards from the midline. The basisphenoid forms the anterior margin of the broad median depression that characterizes the ventral surface of the basioccipital. The µCT scans indicate that the internal carotid artery entered the cranium through a transversely oriented sulcus that is formed by the quadrate and prootic (figure 6a,b). This sulcus is medially floored by the basisphenoid to form the canalis caroticus internus, which continues medially along the suture of the prootic with the basisphenoid. The canalis caroticus internus bifurcates within the sutural contact of the basisphenoid and prootic, and two subordinate canals emerge from this point. Medially, one of these canals punctures the basisphenoid to form the cerebral carotid canal, which curves anteriorly to exit along the lateral walls of the sella turcica, in the portion dorsally covered by the dorsum sellae. The second canal is anteriorly directed, and minimally contains the vidian branch of the facial nerve, but possibly also a palatine artery (see *Pterygoid* above). The canal of the abducens nerve penetrates the basisphenoid posterior to the clinoid process and exits anteriorly below this process. From the anterior exit foramen for the abducens nerve, there is a distinct sulcus for the anterior course of the abducens nerve that is not found in other turtles that we examined (electronic supplementary material. figure S1f). This sulcus is imprinted into the most lateral aspect of the surface of the basisphenoid and parallels the basisphenoid-pterygoid suture along more than half the length of the rostrum basisphenoidale. The medial margin of the sulcus is dorsally expanded to form a low ridge that marks the border between the abducens sulcus and sella turcica. The latter structure is posteriorly expanded to form a deep fossa, which is dorsally overhung by a very low dorsum sellae. The clinoid process is also low and entirely anteriorly directed, with no dorsal inclination. The clinoid process is short and very pointed, and overhangs the posterior part of the abducens sulcus. The rostrum basisphenoidale is broad, anteriorly flattened, and overlies about half of the midline length of the pterygoids.

*Stapes.* Both stapes are partially preserved in UA 10581, but only the right stapes is complete, albeit shattered into four pieces along the stapedial shaft (figure 6f). The two lateral fragments are displaced outwards creating the illusion that a piece is missing in the middle, which is not the case. The stapedial shaft is notably thin and has a circular cross-section. Laterally, the stapedial shaft expands slightly in width and becomes flatter. When the four pieces of the stapes are aligned, the stapes terminates laterally approximately halfway along the mediolateral width of the cavum tympani, indicating that a cartilaginous extrastapes was present, as is common among turtles [56]. The lateral end of the right stapes is still nearly articulated with the fenestra ovalis, and is only somewhat medially displaced from its margin. The medial end of the stapes is expanded to form an oval footplate, which is concave on its medial surface, giving the stapes its characteristic trumpet shape.

*Labyrinth.* As in all turtles, the labyrinth of UA 10581 (figure 6c–f) is housed in a series of cavities and canals that are enclosed by the prootic anteriorly, the opisthotic posteriorly and the supraoccipital dorsally [37]. As in all pleurodires [52], the LSC is not only formed by the opisthotic but has a large contribution from the prootic. A highly unusual feature of UA 10581 is that the quadrate forms a small portion of the lateral wall of the LSC. The fenestra ovalis is open ventrally towards the basicranial floor, as the prootic and opisthotic have no contact in the ventral margin of the fenestra. The labyrinth has a strongly asymmetrical semicircular canal system in which the anterior semicircular canal (ASC) is longer than the posterior semicircular canal (PSC; figure 6d), which is common, but not ubiquitously observed in turtle labyrinths [50,57]. The angle between the ASC and PSC approximates 90°, but the ASC accounts for a larger proportion of this angle when an axial plane is projected through the position of the common crus (figure 6d). This condition can be seen in some turtles for which labyrinths have been reported (e.g. [53]), but the distribution of this feature is currently unclear. Both vertical semicircular

canals (i.e. the ASC and PSC) of UA 10581 are relatively low dorsoventrally compared with those of most turtles (figure 6c,e; [50]). The vertical semicircular canals meet in a dorsally low but broad common crus, which is dorsally only weakly embayed between the two. The vertical semicircular canals are elongate, and both extend significantly anteriorly and posteriorly beyond the extent of the LSC, respectively. The ampullae of the ASC and LSC are distinct structures, particularly because the ASC extends anteriorly far beyond the level of the lateral ampulla. The elongate but low morphology results in a low aspect ratio of the labyrinth, which could be related to the relatively flat nature of the cranium. This suggests that labyrinth shape in turtles may in part be influenced by spatial constraints imposed by braincase shape (see also [58,59]). Despite their lengths, the ASC and PSC are near straight along most of their extent, and most of the canal curvature is concentrated near the ampullae and the common crus (figure 6c). The LSC of UA 10581 differs in this regard from the vertical canals, as it is strongly curved, and forms a wide arc around the lateral perimeter of the vestibule (figure 6d). Highly curved and elongate LSCs are common in pleurodires [60], but the strong width of the LSC in UA 10581 is highlighted by the quadrate contribution to the LSC. The bony bar that separates the lateral surface of the vestibule from the medial wall of the LSC is crescentic in shape and extends posteriorly far around the vestibule, marking the path of the LSC in the anterior portion of the secondary common crus. The secondary common crus is the cavity in the posterior portion of the labyrinth in which the LSC merges with the PSC [50]. However, the approximate path of the LSC through this shared cavity can be seen as a weak impression along the posterodorsal surface of the secondary common crus. The semicircular canals are relatively slender and accordingly have small diameters in comparison with most turtles [50]. However, the semicircular canals are somewhat thicker than those of most extant pleurodires [50,57] and described fossil pleurodires [14,43]. The semicircular canal cross-sections are approximately circular, although the LSC is somewhat more oval (dorsoventrally compressed) than the vertical canals, as is the case in most turtles [50]. The vestibular cavity in the center between the semicircular canals is notably expanded dorsally in UA 10581, and forms a gently convex surface. This morphology has been noted for pleurodires previously [60], although only mentioned specifically for chelids. Given that the general skull morphology and likely feeding mode of UA 10581 resembles that of flat-headed chelids (see Discussion below), it should be tested to determine if expanded vestibular cavities are somehow related to either low but broad skull morphologies or specific ecologies. Vestibular expansion is correlated to ecology in snakes [61] and, although variation of this expansion is certainly larger in snakes than in turtles, it remains untested if similar trends can be found for turtles. The ventral part of the inner ear cavity is relatively featureless, as in other turtles. The cochlear duct leaves no osteological correlate, and only the fenestra ovalis can be discerned as a circular opening at the ventral end of the labyrinth model. Posteriorly, the impression of the incompletely ossified fenestra perilymphatica is visible in the labyrinth model (figure 6e).

### 4.2.2. Mandible

The mandible of UA 10581 is preserved almost fully intact and, because it remains articulated with the cranium, the description is based primarily on the three-dimensional model (figure 7). This structure is noteworthy among pelomedusoids by having a narrow symphysis, a poorly developed labial ridge that lacks any type of sinuosity, and by being almost twice as broad as long. This latter feature is accomplished by contracting the area posterior to the coronoid process, as well as its comparatively far-forward positioned articulation surface on the quadrate, which shortens the overall distance to the tip of the snout. The bones of the mandible are fused near the mandibular articular surface, which suggests that UA 10581 is a skeletally mature individual.

*Dentary.* The majority of the mandible is formed by the dentaries in UA 10581, which are fused to a single unit along the midline (figure 7). The combined fused dentaries form a nearly perfect semicircle. The dentary reveals three distinct, extremely low, parallel ridges on its internal surface. The most external ridge is the labial ridge, which is barely raised above the level of the triturating surface. The second ridge defines the inner margin of the triturating surfaces, which are narrow and flat and only slightly reduced in width near the midline. The third ridge delimits the dorsal margin of the relatively shallow sulcus cartilaginis Meckelii, which extends along nearly the entire internal surface of the mandible, but disappears towards the symphysis. The dentary rises posteriorly to form the coronoid process, which is supported posteromedially by the surangular and anteromedially by the coronoid. In medial view, the dentary contacts the coronoid and prearticular above the sulcus cartilaginis Meckelii, the angular below the sulcus and the surangular posteriorly within the sulcus. The foramen alveolaris inferius is located within the latter contact. A broad posterior contact is present with the surangular and a

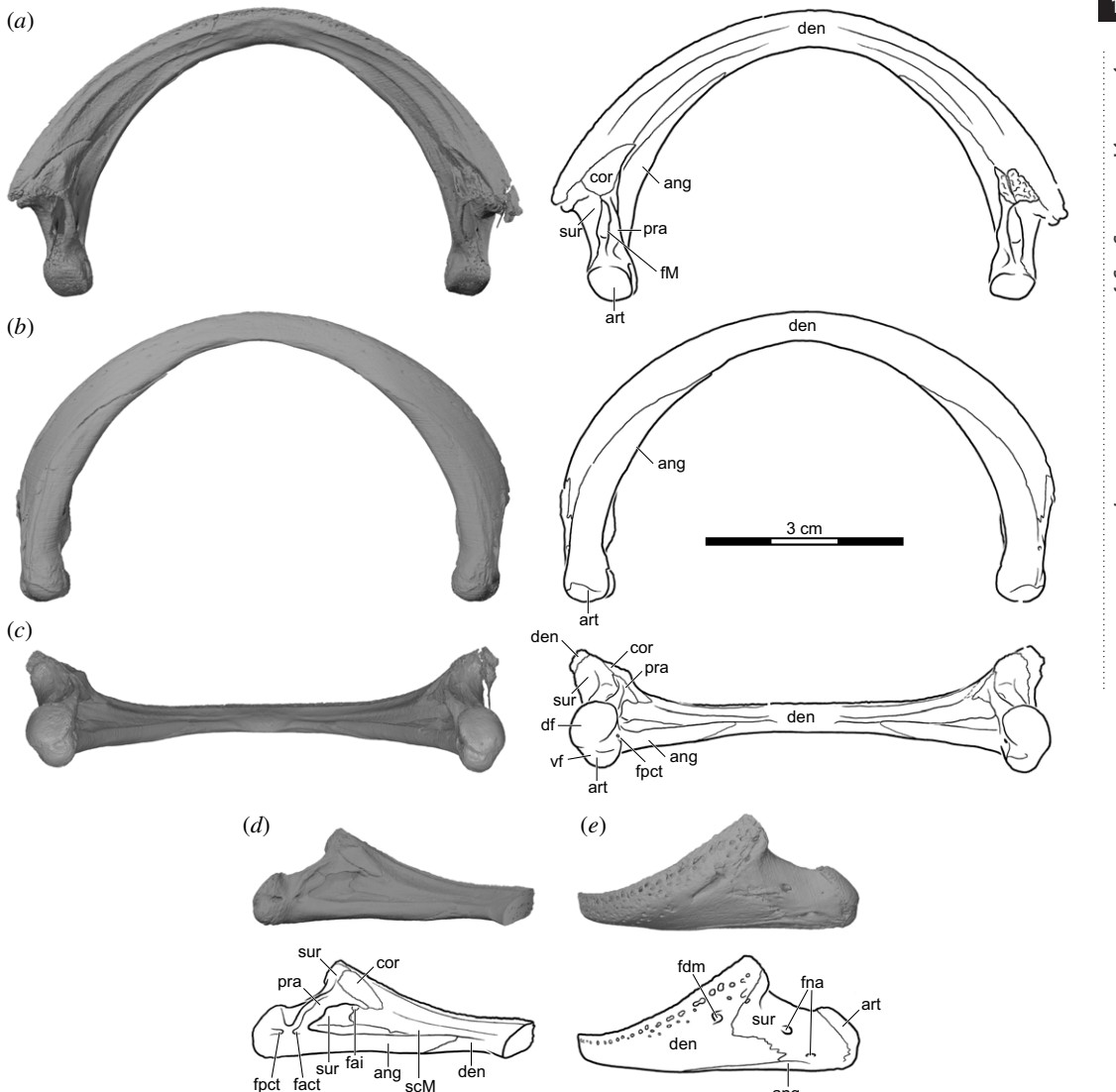

**Figure 7.** *Sahonachelys mailakavava* gen. et sp. nov., UA 10581, holotype, Late Cretaceous (Maastrichtian), Mahajanga Basin, Maevarano Formation, Madagascar. Three-dimensionally rendered model and line drawing of mandible in (*a*) dorsal, (*b*) ventral, (*c*) posterior, (*d*) right medial and (*e*) left lateral views. ang, angular; art, articular; cor, coronoid; den, dentary; df, dorsal facet of articular condyle; fact, foramen anterius chorda tympani; fai, foramen alveolare inferius; fdm, foramen dentofaciale majus; fM, fossa Meckelii; fna, foramen nervi auriculotemporalis; fpct, foramen posterius chorda tympani; pra, prearticular; scM, sulcus cartilaginis Meckelii; sur, surangular; vf, ventral facet of articular condyle.

ventral contact with the angular, which extends along the ventromedial margin of the dentary for most of the length of the mandibular ramus. A series of nutritive foramina are located just below the labial margin on the exterior surface of the dentary. These foramina are densely packed dorsally, and become scarcer more ventrally. The foramen dentofaciale majus is relatively small, is located on the lateral dentary surface shortly anterior to the suture with the surangular, and is ventral to the dense row of neurovascular foramina. The foramen is posteriorly elongated by the presence of a short sulcus, which is unnaturally wide on the left side, making it unclear whether this condition is the result of development or taphonomy. Internally, the foramen dentofaciale majus leads into a canal in each dentary ramus, which continues anteriorly and joins the opposite canal in the symphyseal area of the fused dentaries. This suggests that the rami were already fused at birth, as turtle with unfused dentaries lack communication of blood and nerve canals between both rami (SWE, personal observation).

*Angular.* The angular of UA 10581 forms much of the posteroventral aspects of each mandibular ramus, where it contacts the dentary anteriorly, the articular posterodorsally, the surangular dorsolaterally and the prearticular dorsomedially (figure 7). The angular caps these bones from

ventral. The ventral surface of the angular is transversely convex and wraps around the more dorsally positioned bones of the mandible. The angular tapers anteriorly and develops a pointed process that lies against the ventromedial surface of the dentary. Most of the medial surface of the angular is crossed by an anteroposteriorly oriented ridge that forms the ventral border of the fossa Meckelii and sulcus cartilaginis Meckelii. Within the fossa Meckelii, the angular has a low, dorsal lamina that lies against the medial surface of the surangular. The sutures with the prearticular and articular posterior to the sulcus cartilaginis Meckelii are fused on both sides. We are therefore not able to comment on the location of these contacts. However, in the region of contact with the prearticular, there is a short, anteroposteriorly oriented canal on the medial surface, which forms two foramina. The identity of these foramina is unclear, but they may correspond to the foramen anterius and foramen posterius chorda tympani [37].

*Surangular.* The surangular of UA 10581 forms the posterolateral region of the mandibular ramus (figure 7). The surangular forms the lateral margin of the fossa Meckelii and has an anterodorsal process that buttresses the coronoid process. Along this process, the surangular contacts the dentary anteriorly and the coronoid and prearticular medially. The anterodorsal process of the surangular is thickened dorsally, while the remainder is a relatively thin, vertical sheet of bone. An anteromedial lamina of the surangular within the fossa Meckelii is absent, as reported for other pleurodires [52]. Two foramina are present on the lateral surface of the surangular, which lead to a common groove that communicates with a cavity within the surangular. We interpret both foramina as pertaining to the auriculotemporal nerve. The surangular is largely hidden from view by the prearticular in medial view; however, it is apparent that it forms much of the medial wall of the fossa Meckelii. The foramen alveolare inferius is found at the surangular-dentary contact within the fossa Meckelii. Additional foramina are located on the medial side of the bone, lateral to the prearticular and at the base of the posterodorsal process, that communicate with the inside of the bone and the auriculotemporal nerve foramina. We are unclear about the homology of these foramina.

*Coronoid.* The right coronoid of UA 10581 disarticulated from the mandible and is missing, but the left coronoid remains in place (figure 7). The coronoid is a relatively flat, triangular bone that sits medial to the coronoid process and does not contribute to the triturating surface. Minor crenulations on its dorsal surface suggest that the adductor jaw muscles at least partially inserted on this bone. In medial view, the coronoid contacts the dentary anterodorsally, forms part of the dorsal margin of the sulcus cartilaginis Meckelii anteroventrally, contacts the prearticular posteroventrally and contacts the surangular posterodorsally. A contact with the angular is absent. A contribution to the fossa Meckelii is precluded by a short surangular-prearticular contact.

*Articular.* The articular of UA 10581 forms the posterior margin of the fossa Meckelii and contacts the surangular anterolaterally (figure 7). The anteromedial contacts with the angular and prearticular are obscured due to fusion of these elements with the articular. A short canal is developed in this region, which we interpret as the canalis chorda tympani. The area articularis mandibularis has two parts, which form distinct subfacets. This is best seen in posterior view (figure 7c). These subfacets are developed as two dorsoventrally aligned condyles, of which the dorsal condyle is larger than the ventral condyle. Both surfaces are laterally confluent and medially notched by a sulcus. The dorsal subfacet is clearly formed by the articular, without further contributions by either the surangular or prearticular. This dorsal surface is gently convex and rounded laterally, but bears a shallow longitudinal sulcus more medially and posteriorly. The ventral portion of the area articularis mandibularis is more strongly convex than the dorsal subfacet. This surface is positioned in the area where a retroarticular process is present in some turtles, and may be at least partially formed by the angular. However, the angular-articular suture is unclear externally and in the μCT scans. The ventral subfacet is ventrally bordered by a shallow, ridged margin. This margin separates the smooth surface area of the subfacet from a surrounding area that is gently roughened by small longitudinal striations. This indicates that the jaw extensor musculature inserted ventrally on the mandible, and anterior to the ventral subfacet. As the cranium and mandible of UA 10581 are articulated in their original position (figure 3), the relationships between the area articularis mandibularis and the respective receiving surface on the quadrate can be compared. The mandibular articulation on the quadrate is strongly anteroventrally oriented due to the inclination of the respective process of the quadrate. The surface is furthermore centrally incised by an anteroposterior sulcus that separates medial and lateral subsurfaces of the quadrate. The largely convex dorsal subfacet of the area articularis mandibularis fits neatly into this sulcus. The relationships between both surfaces suggest that the mandible could slide along the sulcus propalinally. As preserved, the mandible is in a forward-slid position. From this position, the ventral subsurface of the area articularis mandibularis would serve as the surface along

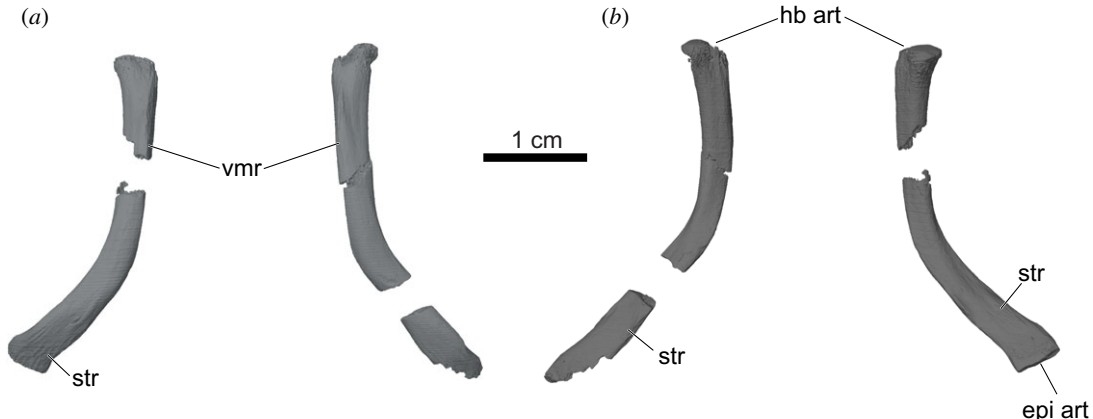

**Figure 8.** *Sahonachelys mailakavava* gen. et sp. nov., UA 10581, holotype, Late Cretaceous (Maastrichtian), Mahajanga Basin, Maevarano Formation, Madagascar. Three-dimensional renderings of ceratobranchials I in (*a*) ventral and (*b*) dorsal views. epi art, epibranchial articulation surface; hb art, hyoid body articulation surface; str, striated bone texture; vmr, ventromedial ridge of ceratobranchial I.

which the mandible is rotated when the lower jaw is opened. This may have functional significance (see Discussion below).

*Prearticular*. The prearticular of UA 10581 forms the medial margin of the fossa Meckelii. It has a short lateral contact with the surangular anterior to the fossa Meckelii and a broad anterior contact with the coronoid dorsal to the fossa Meckelii (figure 7). The posterior contacts with the articular and surangular are obscured by fusion of these elements. In comparison with other turtles, the anteroposterior width of the prearticular is extremely reduced, giving it a rod-like appearance. As a result, the foramen intermandibularis medius, which is located at the posterior margin of the sulcus cartilaginis Meckelii, is pushed posterior to the level of the coronoid process.

*Splenial*. Given the exceptional preservation in UA 10581, we are confident that a splenial is absent. This is the typical condition for pelomedusoids [5].

### 4.2.3. Hyoid apparatus

The first pair of ceratobranchials (= ceratobranchial I, ceratohyal I, cornu branchial I) is preserved on the underside of the cranium of UA 10581 (figure 3). As the specimen is otherwise preserved in great detail, the absence of additional hyoid bones (e.g. ceratohyals II, ceratobranchials II, hyoid body) is probably due to the lack of original ossification of these elements, rather than a preservational artefact [62].

Ceratobranchial I is an elongate, recurved element (figure 8). The anterior portions of both left and right elements lie parallel to one another and ventrally cover parts of the palate, but the posterior portions wrap laterally behind the articular process of the quadrate. The anterior surface of ceratobranchial I forms a well-developed facet for articulation with the processus medialis of the hyoid body. The articular surface is gently convex, which is otherwise only known to occur in pelomedusids among extant turtles [62]. At mid-length, the element is oval in cross-section, as seen along some breaks in both elements, whereby it is broader mediolaterally than dorsoventrally. A low but distinct ridge is developed on the ventromedial surface of ceratobranchial I, indicating the insertion of musculature. At two-thirds of its length, ceratobranchial I becomes notably flatter dorsoventrally, and also broadens somewhat mediolaterally. The peripheral surfaces around the distal end are striated by low anteroposterior furrows. The posterior surface, which is preserved on the right side, shows the development of a concave articular facet, providing evidence for the former presence of an epibranchial element that was probably cartilaginous.

The curvature and position of the ceratobranchial Is of UA 10581 match well with the morphology and position of these elements in extant turtles. This suggests that the elements are broadly preserved in their resting position, although minor shifting must have occurred due to the decomposition of the tissues surrounding the neck and mouth. The anterior portions of the ceratobranchials are sagittally oriented and not turned medially towards the hyoid body, which may have implications regarding the feeding ecology of this animal (see Discussion below).

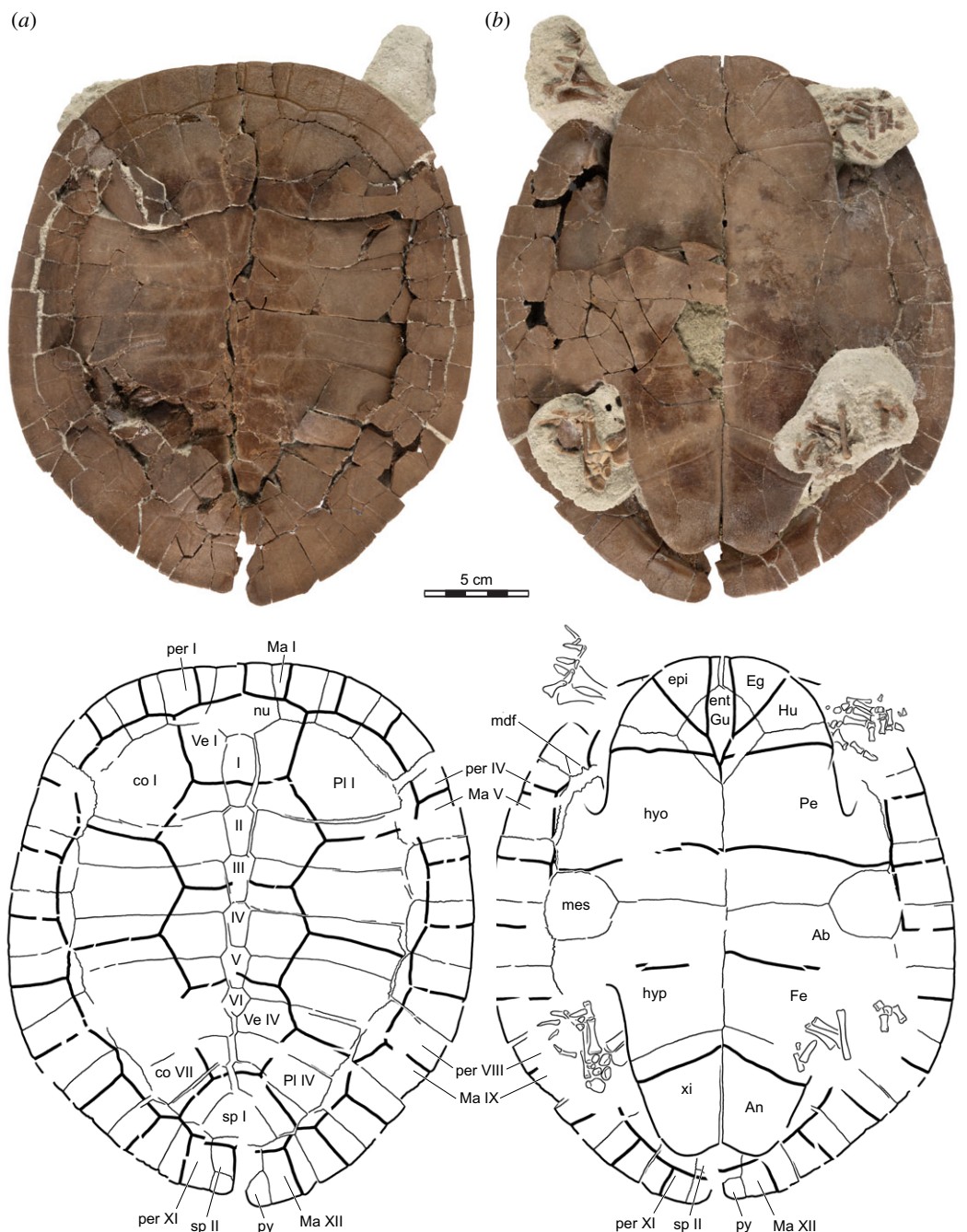

**Figure 9.** *Sahonachelys mailakavava* gen. et sp. nov., UA 10581, holotype, Late Cretaceous (Maastrichtian), Mahajanga Basin, Maevarano Formation, Madagascar. Photograph and line drawing of shell in (*a*) dorsal and (*b*) ventral views. The lone Roman numerals indicate the neurals. Ab, abdominal scute; An, Anal scute; co, costal; Eg, extragular scute; ent, entoplastron; epi, epiplastron; Fe, femoral scute; Gu, gular scute; Hu, humeral scute; hyo, hyoplastron; hyp, hypoplastron; mdf, musk duct foramen; Ma, marginal scute; mes, mesoplastron; nu, nuchal; Pe, pectoral scute; per, peripheral; Pl, pleural; py, pygal; sp, suprapygal; Ve, vertebral scute; xi, xiphiplastron.

### 4.2.4. Shell

The shell of UA 10581 probably came to rest intact, but suffered minor damage in the form of slight disarticulation, shifting of elements, minor crushing, and breakage and loss of substance during recovery (see Geological setting above). The limbs were left intact as found, thereby blocking full view of the interior of the shell. The vast majority of external features, however, can be observed with confidence, especially as all sutures and sulci are crisply preserved (figure 9).

The carapace is mostly elongate anteroposteriorly and rounded, but tapers slightly towards the posterior end. Only a small, but distinct pygal notch is present. The surface is generally smooth, but decorated with the finely vermiculated pattern typical of pleurodires. The plastron is attached to the carapace by a finely sutured bridge. Plastral fontanelles are absent. The anterior plastral lobe has a broad base, parallel sides at the level of the pectoral scute, a rounded anterior margin, and broadly covers the anterior shell opening. The posterior plastral lobe is narrower at its base compared with that of the anterior plastral lobe, the anterior half has parallel sides, but the posterior half tapers gradually. Although difficult to appreciate from the illustrations (figure 9), a distinct, but shallow anal notch is present. The posterior plastral lobe only partially covers the carapace from below.

*Carapacial bones*. The carapace of UA 10581 consists of six neurals, eight pairs of costals, a nuchal, 11 pairs of peripherals, two suprapygals, and a pygal (figure 9*a*). Its midline length is approximately 25.5 cm. The six relatively narrow neurals form a continuous series of elements that prevent costals I–V from contacting one another along the midline. Neural VI partially prevents costals VI from contacting one another along the midline. Neural I is a four-sided element that is about twice as long as wide. Neurals II–V are hexagonal elements with short anterolateral sides. Neural VI is pentagonal with short anterolateral sides. In general, the neurals have a similar width but gradually decrease in their anteroposterior dimensions towards the posterior. As a general trend, the costals decrease in their dimensions from anterior to posterior. A notable exception is costal I, which is about twice as long anteroposteriorly as costal II, but less broad transversely. The anterior elements are furthermore oriented laterally, while the lateral ends of the posterior elements are increasingly oriented posteriorly. All costals contact two neighbouring peripherals, with the exception of costal I, which contacts four peripherals. The nuchal is a relatively small element that is nearly as wide as long. The 11 pairs of peripherals form a continuous ring that forms the lateral rim of the shell. The posterior limbs block view of the posterior aspects of the bridge, but comparison with other pelomedusoids suggests that the bridge spans from peripherals III to VIII. Peripheral I has a broad posterior contact with costal I. The shell is unusual among pelomedusoids in possessing two suprapygals, not just one [5,6]. Suprapygal I resembles the suprapygal of most other pelomedusoids in being a sub-triangular element with a broad anterolateral contact with costal VIII, having a broad posterior contact with suprapygal II, and having a short posterolateral contact with peripheral XI. Suprapygal II and the pygal in combination resemble the pygal of other pelomedusoids by filling the elongate space between suprapygal I anteriorly and peripherals XI laterally. Future discoveries will clarify if this is a feature typical of this taxon, or a developmental anomaly of this specimen. The pygal forms a distinct but small pygal notch.

*Carapacial scutes*. The carapace of UA 10581 is covered by five vertebrals, four pairs of pleurals, and 12 pairs of marginals (figure 9*a*). Damage to the nuchal creates the illusion that a narrow cervical scute is present as well, but close study confirms the presence of a break. The five vertebrals have similar anteroposterior dimensions. Vertebral I is a heptagonal element that broadens slightly towards its anterior end. It contacts marginal I anteriorly, the medial half of marginal II anterolaterally, pleural I posterolaterally and vertebral II posteriorly. Vertebrals II–IV are hexagonal elements that decrease in width posteriorly and have two lateral contacts each with the pleural series. Anteriorly protruding, rounded midline inflections are developed between these vertebrals. Vertebral V is a heptagonal element that expands posteriorly and that contacts vertebral IV anteriorly, pleural IV anterolaterally, and the medial halves of marginals XI and XII posteriorly. The intervertebral sulci cross neurals I, III, V and costals VIII. The pleurals are nearly as wide as the vertebrals. Pleural I contacts marginals II–V, pleural II contacts marginals V–VII, pleural III contacts marginals VII–IX and pleural IV contacts marginals IX–XI. The marginals form a complete ring of peripheral elements that do not cover the costals or suprapygal I. Marginals IV–VIII medially contact the plastral scutes.

*Plastral bones*. The plastron of UA 10581 consists of an entoplastron and paired epi-, hyo-, meso-, hypo- and xiphiplastra (figure 9*b*). Its midline length is about 23.5 cm. The epiplastron is a large element with a broad midline contact with its counterpart anterior to the entoplastron. The posterior contact with the hyoplastron is nearly transverse. The entoplastron is a large, diamond-shaped element with equally sized anterolateral and posterolateral contacts with the epiplastra and hyoplastra, respectively. As in all other pelomedusoids, the main body of the plastron is formed by the hyo-, meso- and hypoplastron. The hyoplastron forms an axillary buttress that reaches anteriorly to contact peripheral III and that terminates on costal I. An internal view of the shell available during preparation suggests that the axillary buttress does not cross underneath costal II. Two musk duct foramina are developed along the marginal IV-pectoral sulcus at the peripheral III–hyoplastron suture. The mesoplastra are about as long as wide and wedged between the lateral margins of the hyo- and hypoplastra. The extent of the inguinal process cannot be assessed with full confidence, but the location of the inguinal notch together

with damage to the carapace caused by the inguinal buttress being taphonomically pushed upwards is suggestive of contact with peripheral VIII and costal VI. The xiphiplastron is a rectangular element that shows a finely sutured anterolateral contact with the hypoplastron. The posterior margin of each xiphiplastron is transversely convex, so that both elements together define a shallow anal notch.

*Plastral scutes.* The plastron of UA 10581 is covered by a gular scute and paired extragulars, humerals, pectorals, abdominals, femorals and anals. Inframarginals are absent (figure 9b). The gular is a particularly elongated element that extends from the anterior plastral margin posteriorly and almost fully crosses the entoplastron, but does not reach the hyoplastron. The gular posteriorly contacts the pectorals, thereby precluding a midline contact of the extragulars and humerals. The extragulars are triangular elements that have a medial contact with the gular, a posterolateral contact with the humeral and clearly cover the anterolateral aspects of the entoplastron. The humeral-pectoral sulcus is located behind the epiplastron, but clearly crosses the entoplastron. The pectorals, abdominals, femorals and anals contribute nearly equally to the midline sulcus. The pectoral-abdominal sulcus is located just anterior to the mesoplastra. The abdominal-femoral sulcus probably ends laterally within the inguinal notch. The femoral-anal sulcus is located on the xiphiplastron.

### 4.2.5. Vertebral column

The cervical column of UA 10581 is partially preserved, but the identification of all elements is partially hindered by significant erosional damage. We here only figure cervicals I–III, as these are the best-preserved elements (figure 10a,b). We tentatively identify two additional, poorly preserved elements as cervicals IV and VIII, but do not figure them, as they are too damaged to show any meaningful morphology.

The cervicals generally resemble those of other pleurodires in having tall, posteriorly tilted dorsal processes that bear the postzygapophyses and transverse processes that are placed at mid-length. The sole presence of procoelous vertebrae posterior to the axis is consistent with the pelomedusoid pattern of (2))3))4))5))6))7))8). When preserved, the central articulations form regular, rounded surfaces that are about as high as wide. The postzygapophyses are tightly spaced, but never confluent along the midline. The arches of the atlas are not sutured to one another along the midline, nor is its intercentrum sutured to its centrum. The postzygapophyses of the atlas face ventromedially. The axis has a well-developed anterior process on its dorsal surface. We cannot comment on the central surfaces of cervical VIII, but a straight keel is present on its ventral side. The midline length of cervical II is approximately 18 mm, that of the atlas is 10 mm. Assuming that cervicals III–VIII had similar dimensions, we calculate a neck length of approximately 13.6 cm. This length, combined with the midline length of the carapace (25.5 cm), gives a neck to carapace ratio of about 53%, which, keeping the imprecision of our measurement in mind, is broadly comparable to that of the box turtle *Terrapene carolina* (53%), the alligator snapping turtle *Macrochelys temminckii* (53%), the tortoise *Kinixys belliana* (54%), and only slightly shorter than the 'long-necked chelid' *Chelus fimbriatus* (59%, see electronic supplementary material).

### 4.2.6. Girdles and limbs

The pectoral girdle of UA 10581 is preserved in the form of a partial scapulocoracoid (figure 10c,d). It is not possible to clarify if this is the right or left element, as the two scapular processes are damaged. We are nevertheless able to conclude that the coracoid is a strap-like element, that the scapula possesses a distinct neck, and that the angle between the two scapular processes is less than 70°. The pelvis is fully blocked from view, with the exception of the sutural contact of the ilium with the carapace, which reaches costal VII but none of the pygal elements.

The right forelimb of UA 10581 is preserved with the manus exposed in dorsal view and the humerus covered by the shell (figure 10e). Four claws are visible. The smallest claw is too large to fit digit V, at least as preserved on the left side, so we identify them as belonging to digits I–IV. The left manus is also exposed in dorsal view (figure 10f), the remainder of the limb being hidden by matrix. All five metacarpals are present. As in other turtles, metacarpal I is the shortest and most robust. The remaining metacarpals are notably slim, especially metacarpal V. The only digit preserved in articulation is digit I, which consists of two phalanges. Nine elements, of which three are claws, are associated with metacarpals II–IV. The manual formula is therefore probably 2-3-3-3-?. The first phalanx of these digits is notably shorter than the second phalanx. A number of carpals are present as well, but we cannot assign them to a particular position.

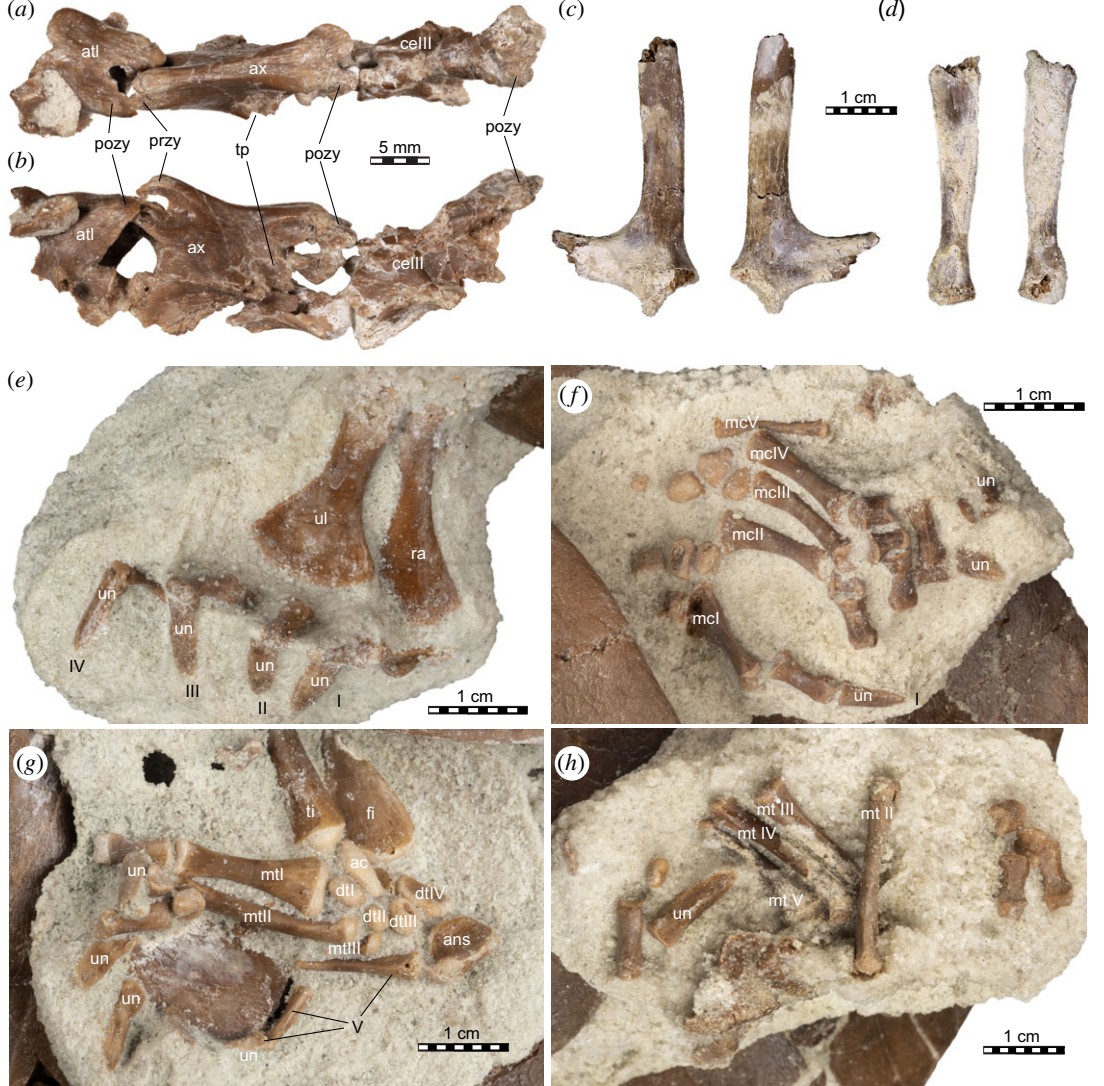

**Figure 10.** *Sahonachelys mailakavava* gen. et sp. nov., UA 10581, holotype, Late Cretaceous (Maastrichtian), Mahajanga Basin, Maevarano Formation, Madagascar. Photographs of cervical vertebrae I–III in (*a*) dorsal and (*b*) left lateral views. Photographs of (*c*) scapula and (*d*) coracoid. Damage to both obscures the views. Photographs of (*e*) right forelimb in dorsal view, (*f*) left forelimb in dorsal view, (*g*) right hind limb in plantar view and (*h*) left hind limb in plantar view. Roman numerals indicate digit number. ans, ansula; atl, atlas; ax, axis; ce, cervical vertebra; dt, distal tarsal; fi, fibula; mc, metacarpal; mt, metatarsal; pozy, postzygapophysis; przy, prezygapophysis; ra, radius; ti, tibia; tp, transverse process; ul, ulna; un, ungual.

The right hind limb of UA 10581 in preserved with the crus and pes exposed in plantar view, while the femur remains hidden within the shell (figure 10*g*). The distal ends of the tibia and fibula are exposed. The astragalocalcaneum is the largest element in the middle foot, but is mostly hidden from view. Four distal tarsals are present, with the fourth tarsal being the largest. The ansula (i.e. the fused distal tarsal V and metatarsal V [63]) is a large blocky element with a distinct hook. Three elements connect to the ansula, which we identify as the slim fifth digit consisting of a metatarsal-shaped first phalanx and two short distal phalanges. The claw is reduced. The remaining parts of the pes consist of three partially to fully exposed metatarsals and seven phalanges, of which three are claws. The preserved portions of the left pes consist of metatarsals II–IV, the elongate first phalanx of digit V, and seven phalanges, of which one is a claw (figure 10*h*). The well-developed articular surfaces, in combination with the pose in which the specimen is preserved, suggest that the manus and pes were highly mobile.

## 4.3. Phylogeny

The phylogenetic analysis resulted in 11 most parsimonious trees with a best score of 55.63475. The resulting topology implies 1298 steps under equal weight of 1. An extract of the strict consensus tree

is provided in figure 2. The full strict consensus tree and the full list of common synapomorphies are provided in the electronic supplementary material. The strict consensus topology suggests that *Sahonachelys mailakavava* gen. et sp. nov. and *Sokatra antitra* form the clade *Sahonachelyidae*, herein defined as the maximum inclusive fossil clade that includes *Sahonachelys mailakavava*. This clade is the immediate sister of an unnamed clade formed by *Podocnemidoidae* and *Bothremydidae*. The successive pan-pelomedusoid outgroups are *Atolchelys lepida*, *Pan-Pelomedusidae* and *Araripemydidae*.

# 5. Discussion

## 5.1. Alpha taxonomy and phylogenetic relationships

A clear majority of fossil pan-pelomedusoids are known from shells or skulls alone (see [5,6] for summaries) making it difficult to rigorously establish new species. The rare preservation of a nearly complete skeleton from the Late Cretaceous Maevarano Formation of Madagascar that includes both anatomical systems therefore allows us to establish *Sahonachelys mailakavava* as a new taxon with confidence. As the vast majority of characters that help differentiate this species also have a phylogenetic signal, we discuss both aspects in parallel below.

Many fossil pan-pelomedusoids are known from shell material only [5,6]. In general, pan-pelomedusoids have low-domed, oval shells that lack a cervical scute; show a reduced number of neurals, which allows the posterior costals to contact one another along their midline; have well-developed buttresses; and have broad plastra with equidimensional mesoplastra [5,6]. *Sahonachelys mailakavava* broadly conforms to this pattern but also has a highly unusual arrangement of the scutes along the anterior plastral row. Namely, the extragulars are large, triangular elements that broadly contribute to the anterior plastral margin and clearly cross onto the entoplastron, while the gular scute is an extremely elongate element that narrowly contributes to the anterior plastral row and nearly crosses the entoplastron to reach the hyoplastra. Among pan-pelomedusoids, the resulting lack of a midline contact of the humerals is homoplastically found in the taphrosphyines *Taphrosphys sulcatus* and *Ummulisani rutgersensis* [5] and the kurmademydine *Jainemys pisdurensis* [19], but particularly broad exposure of the extragulars along the anterior margin of the plastron combined with the particularly narrow exposure of the gular is uniquely developed in *Sahonachelys mailakavava*. Future discoveries will clarify if this arrangement is found to this degree in its immediate sister taxon, *Sokatra antitra*, which is only known from its cranium [34]. *Sahonachelys mailakavava* is easily distinguished from the only named Late Cretaceous shell-based pelomedusoid from Madagascar, the Cenomanian *Akoranemys madagasika* [21], by having proportionally wider vertebral scutes that laterally taper to a point and proportionally smaller mesoplastra that are not crossed by the pectoral-abdominal sulcus. The shell of *Sahonachelys mailakavava* uniquely exhibits two pygal elements between the last pair of peripherals but, as no other crown pleurodire exhibits this condition, it is plausible that this is a congenital abnormality.

A number of cranial synapomorphies place *Sahonachelys mailakavava* gen. et sp. nov. within the clade *Pan-Pelomedusoides*, including the absence of nasals, the presence of a midline between the prefrontals, a deep upper temporal emargination, and the absence of a splenial. Within *Pan-Pelomedusoides*, *Sahonachelys mailakavava* is united with the clades *Podocnemidoidae* and *Bothremydidae* relative to *Atolchelys lepida*, *Pan-Pelomedusidae* and *Araripemydidae* by the exclusion of the basioccipital from the occipital condyle, a broad basioccipital-quadrate contact that floors the processus interfenestralis, the presence of a pterygoid flange that does not reach the quadrate ramus of the pterygoid, the presence of an exoccipital-quadrate contact, the presence of a very short basioccipital, and a reduced ventral exposure of the prootic in ventral view. On the other hand, *Sahonachelys mailakavava* is excluded from the unnamed clade formed by *Podocnemidoidae* and *Bothremydidae* by plesiomorphically possessing a dorsoventrally flattened cranium, a foramen palatinum posterius that is located in a flat surface relative to the triturating ridge, parallel-sided triturating surfaces, a minor contribution of the palatine to the triturating surfaces, by lacking a cavum pterygoidei or fossa pterygoidea, and the presence of a reduced ventral exposure of the prootic. The above-listed characters allow differentiation of *Sahonachelys mailakavava* from all named pan-pelomedusids except *Sokatra antitra*, including the coeval *Kinkonychelys rogersi* [33] and the unnamed podocnemidoid [35] from the Maevarano Formation.

The immediate sister group relationship of *Sahonachelys mailakavava* gen. et sp. nov. with *Sokatra antitra* is supported by a reduced contribution of the maxilla to the floor of the orbit (homoplastically developed in some large-bodied taphrosphyines), and, most apparent, the presence of distinct posterior process of the maxilla. These two turtles are the only known representatives of the clade

*Sahonachelyidae. Sahonachelys mailakavava* differs from *Sokatra antitra* by having a much flatter and broader cranium with more dorsally oriented orbits, the presence of a prefrontal-palatine contact, the absence of parietal-palatine contact, by having narrower triturating surfaces that lack a lingual ridge, a narrower midline contact of the palatines, the presence of a distinct supramaxillary artery sulcus on the ventral side of the jugal, and the exposure of the prootic anterior and posterior to the foramen posterius canalis carotici interni.

The µCT scan data suggest a number of unusual characteristics in the cranium of *Sahonachelys mailakavava*, but the lack of comparative data, particularly for *Sokatra antitra*, makes it impossible to determine if these represent autapomorphies, synapomorphies, or individual variations. These morphological features include the presence of a small, slit-like posterior opening to the antrum postoticum between the squamosal and quadrate, a quadrate contribution to the trigeminal foramen, a quadrate contribution to the LSC, a foramen in the floor of the sulcus cavernosus that communicates with the facial and/or carotid canal system, an abducens sulcus on the lateral margin of the rostrum basisphenoidale, an extremely voluminous recessus scalae tympani that is possibly associated with the absence of a glossopharyngeal canal through the opisthotic, and the ventral entry of the supramaxillary artery.

Our phylogenetic hypothesis broadly corresponds with previous ones [3,34] by hypothesizing that *Sahonachelyidae* is sister to the clade formed by *Podocnemidoidae* and *Bothremydidae*. This is not particularly surprising, as we used a modified version of the matrix assembled by [3], which in turn is a variant of the matrix used by [34]. *Sahonachelys mailakavava*, therefore, does not appear to be a tree-changing taxon, but rather hints at a growing stability in the understanding of pelomedusoid relationships.

We note two interesting implications for this topology. First, the oldest known representatives of the clade formed by *Podocnemidoidae* and *Bothremydidae* are known from the late Early Cretaceous (e.g. the Albian *Brasilemys josai* and *Cearachelys placidoi*). This implies a sahonachelyid ghost lineage that extends from at least the Albian to Campanian, a time span of nearly 45 Myr, a fact already alluded to by [34] and consistent with phylogenetic assessments for various other vertebrate taxa from the Late Cretaceous of Madagascar (e.g. [25,32,64,65]). Second, although sampling of vertebrates from the pre-Maastrichtian Late Cretaceous of both Madagascar and the Indian subcontinent is still very limited, it is notable that no sahonachelyids have yet been found on the latter in the reasonably well-sampled Maastrichtian [66]. Given that the distribution of continental pleurodires seems to contain a strong biogeographic signal [67], a literal interpretation of the topology obtained herein suggests that Madagascar split from the rest of Gondwana as early as the late Early Cretaceous. As Madagascar is usually hypothesized to have been connected with the Indian subcontinent until the Coniacian, at approximately 88 Ma [68–77], however, the conspicuous absence of sahonachelyids on the subcontinent in the Maastrichtian may be the result of extinction.

## 5.2. New insights regarding the morphology of *Sokatra antitra*

Sahonachelyids display an overall unusual morphology relative to other extinct pan-pelomedusoides, particularly representatives of the speciose clade formed by *Podocnemidoidae* and *Bothremydidae*. The holotype of *Sokatra antitra*, a cranium, is relatively complete, but poor preservation obscured a few important morphological details. The discovery of the exceptionally preserved holotype of *Sahonachelys mailakavava* gen. et sp. nov. affords us the opportunity to comment on, and re-evaluate, select characters that were initially not documented or interpreted incorrectly.

Although posterior processes of the maxilla are apparent in the photographs and line drawings of *Sokatra antitra* provided in the type description [34], these structures are not mentioned in the description and are absent from the reconstruction. As pronounced posterior processes are otherwise not present to such a degree in pleurodires, we speculate that the slit-like anterior expansion of the lower temporal emargination may have been interpreted as a break. This character serves as the most conspicuous synapomorphy of the clade *Sahonachelyidae*.

A second character related to the lower temporal emargination is the absence of the ventral process of the quadratojugal. In the vast majority of turtles, the quadratojugal forms a ventral process that extends along the anterior margin of the cavum tympani to nearly reach the mandibular condyle [37]. Gaffney & Krause [34] suggested that the ventral process is broken in the holotype of *Sokatra antitra*, but the excellent preservation of the quadratojugal in the cranium of *Sahonachelys mailakavava* suggests instead that the ventral process was absent. The ventral process of the quadratojugal is also absent in a broad selection of other pelomedusoids [5], but a phylogenetic signal is not apparent.

Gaffney & Krause [34] reported that the incisura columella auris in *Sokatra antitra* is separated from the fenestra postotica by a short wall. The incisura is therefore posteriorly enclosed. However, the

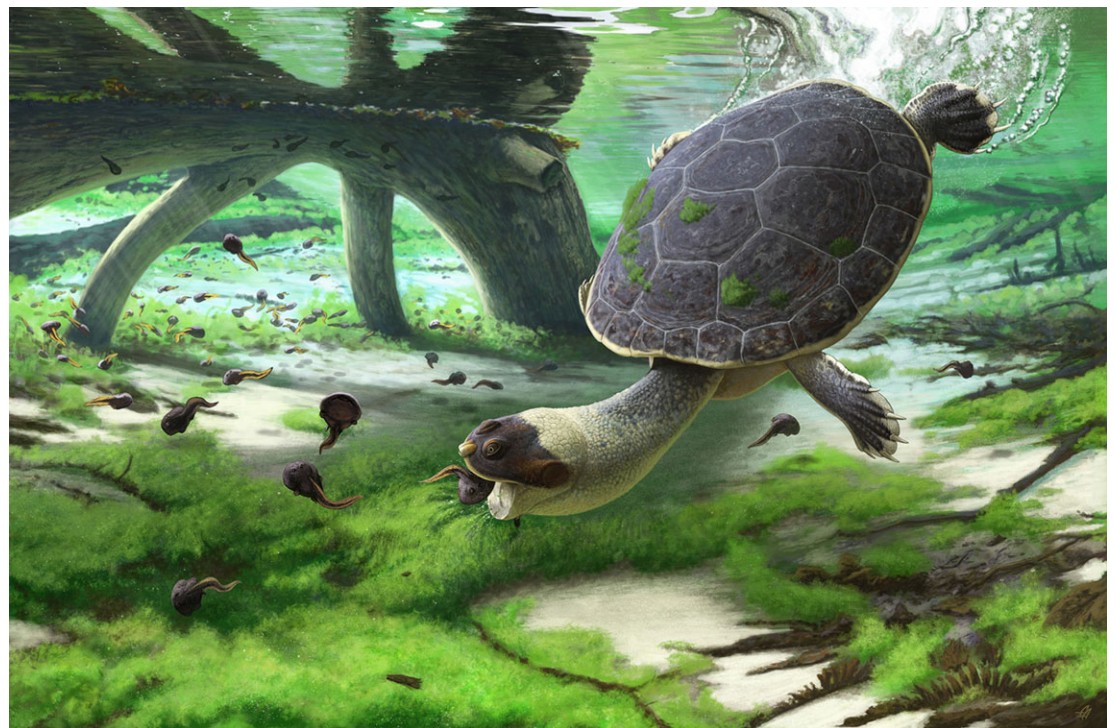

**Figure 11.** A reconstruction of *Sahonachelys mailakavava* gen. et sp. nov. preying upon young larvae of the giant Madagascan frog *Beelzebufo ampinga* using specialized suction feeding. Artwork by Andrey Atuchin.

resulting canal, according to Gaffney & Krause [34], contains both the stapes and the Eustachian tube, as found in podocnemidids, and not the stapes alone, as found in bothremydids. A distinct notch next to the incisura columella auris of *Sahonachelys mailakavava* suggests to us that the Eustachian tube was separated from the stapes by a bony wall. Our analysis of the holotype specimen (UA 9769), as well as all referred material, of *Sokatra antitra* suggests a similar morphology for this taxon as well.

A final, more speculative comment pertains to the possible presence of a vestigial vomer in *Sokatra antitra*. The palatines were originally suggested to form a median process that protrudes anteriorly into the internal choanae [34]. The palatines are more poorly developed in *Sahonachelys mailakavava* and it is therefore perhaps not surprising that the vomer is absent in this taxon. A closer look at the median process in the holotype of *Sokatra antitra*, however, reveals symmetric lines that could be interpreted as sutures defining the posterior contacts of a vestigial vomer. The use of µCT scans may provide additional insights into this morphological question.

## 5.3. Palaeoecology

Although most turtles are dietary generalists, many turtles show dietary specializations, including herbivory, carnivory, durophagy, medusivory or piscivory [78]. Initial studies using geometric morphometrics suggest that turtles with particular specializations do not converge upon the same global skull shape [22]. However, the qualitative analysis of cranial characteristics still allows for the identification of feeding specializations with ease [22].

*Sahonachelys mailakavava* gen. et sp. nov. displays a number of putative morphological adaptations that enable us to identify it as a specialized suction feeder that probably fed on small-bodied prey, such as insect larvae, tadpoles, or fish fry (figure 11). From a theoretical point of view, aquatic vertebrates ingest food items along a spectrum ranging from a pure ram strike, where the head of the predator is thrust forward while the prey item remains stationary, to a pure suction strike, where only the prey moves while the predator remains stationary [79]. In the case of turtles, some form of suction is involved in both types of feeding, as turtles with a ram strike must use compensatory suction to counteract the bow wave that is created in front of the thrusting head [79,80,81]. Ram feeding in turtles allows for the initial apprehension (i.e. grasping) of food items, which can then be orally processed and swallowed using powerful jaws and the tongue [82,83]. Suction feeding in turtles, on

the other hand, uses inertial suction to draw prey items into the oral cavity. However, as a ram component is still present, even in the most specialized suction feeding turtles, a certain portion of suction is used to compensate for the bow wave [80]. The weight of the mandible and the size of the tongue is reduced in turtles that employ suction feeding [79]. However, the reduction of these elements also impedes their ability to manipulate and transport food items within the oral cavity once prey items have been captured. Specialized suction feeders therefore tend to ingest prey items directly into their oesophagus [79], which obviates the need for well-developed triturating surfaces and strong jaw-closing musculature [82]. Early studies extrapolated existing ecological data to suggest, broadly, that cryptodires are ram feeders and pleurodires are suction feeders [81]. However, we note that ram feeding also occurs in pleurodires [84] and suction feeding in cryptodires [85,86].

Although much plasticity remains, specialized suction-feeding turtles (i.e. animals that primarily rely on inertial suction to feed) display a series of characteristic morphological adaptations. All of these adaptations are found, in various degrees, in *Sahonachelys mailakavava*.

Extant suction-feeding turtles have low, hydrodynamic skulls that reduce the production of bow waves [79,87]. In this regard, the skull shape of *Sahonachelys mailakavava* most closely resembles that of *Chelodina* spp. or *Hydromedusa* spp., but is less specialized than *Chelus fimbriatus*, where the skull tapers anteriorly to a point in lateral view [37]. The low skull and gracile jaws of *Sahonachelys mailakavava* suggest that it did not have a powerful bite [82].

The gape of extant suction-feeding turtles is laterally restricted and positioned towards the very front of the animal to help focus suction directed at a prey item [79,88,89]. Several morphological features suggest that both of these adaptations are exaggerated in *Sahonachelys mailakavava*. The cranium and mandible are both extremely short and broad, the mandibular condyles are shifted anteriorly, and the jaws are evenly arched. In addition, the mandible is slightly protractible, which probably would have yielded a wide, near circular gape (figure 11) and facilitated effective fluid flow towards the mouth cavity, a prerequisite for suction feeding in vertebrates [89]. We are unaware of any other turtle with such extreme geometry to the cranium and lower jaws, as other specialized suction-feeding turtles, such as *Chelodina* spp. and *Hydromedusa* spp., have more elongate crania.

Suction is created in turtles by first opening the mouth and then rapidly depressing the hyoid apparatus, which expands the volume of the oral cavity and adjoining oesophagus. Specialized suction feeders possess a broad hyoid apparatus that floors much of the oral cavity in resting position and serves as the attachment site for strong muscles [79,87,88]. The hyoid apparatus of turtles shows great variability in regard to its size and level of ossification [62], and several general ecological trends are apparent. Terrestrial-feeding taxa have a slender hyoid apparatus with thin, poorly ossified branchial arches and a cartilaginous hyoid body [83,87]. Ram-feeding taxa possess a broader hyoid apparatus with thicker, well-ossified branchial arches and a cartilaginous hyoid body [90,91]. Suction-feeding taxa have an enlarged hyoid apparatus with extensive, fully ossified branchial arches and an ossified hyoid body [79,80,85]. As preserved, the hyoid apparatus of *Sahonachelys mailakavava* consists of enlarged and well-ossified first ceratobranchials I (figures 3 and 8). Although the ossification of a single branchial arch is more consistent with ram feeding, we note several other morphological features that are more consistent with suction feeding. The branchial arch is unusually well developed relative to the size of the cranium, ridges protruding along the ceratobranchial shaft indicate the former presence of well-developed hyoid musculature, the anterior portions of the branchial arch are oriented parasagitally, and the articulation site with the hyoid body is convex. The latter two observations are broadly comparable with extant *Chelodina* spp., where the hyoid apparatus is modified into a kinetic structure that unfolds during feeding to greatly and rapidly expand the oral cavity and oesophagus [80]. The absence of a second set of ossified branchial arches may be a preservational artefact, as damage is apparent to the holotype specimen of *Sahonachelys mailakavava* posterior to the skull in the cervical region. However, given the great care invested in the preparation of this specimen, we find it more likely that the absence of a second set of ossified branchial arches is a phylogenetic effect. This hypothesis is supported by the fact that the only other known specialized suction-feeding pelomedusoids, the araripemydids *Araripemys barretoi* and *Taquetochelys decorata*, have only been reported to possess a single pair of ossified ceratobranchials [5,92].

Among living turtles, specialized suction feeding is often associated with a long neck [80]. Our data for the neck length to carapace length ratio in fossil and living turtles shows that this ratio varies broadly across the phylogenetic tree, even when keeping the imprecision of our measurements in mind. The fossil *Ordosemys* sp. has the smallest ratio (25%), while the extant trionychid *Trionyx triunguis* the greatest (105%) (see electronic supplementary material). Interestingly, the most specialized extant suction-feeding turtle, *Chelus fimbriatus*, has only an intermediate neck length to carapace length ratio of 59%.

While no extant suction-feeding turtle has a short neck, numerous extant ram-feeding turtles, particularly trionychids, do have a long neck. Thus, a long neck by itself is indicative of suction feeding, but should not be the only source of evidence when trying to evaluate between suction-feeding versus ram-feeding ecologies. We estimate that the neck length to carapace length ratio in *Sahonachelys mailakavava* was about 53%. Among pleurodires, the neck of this animal is, therefore, more elongate than that of extant pelomedusoids (e.g. *Pelomedusa subrufa*: 49%; *Podocnemis unifilis*: 27–35%), comparable to the extant chelid *Chelus fimbriatus* (59%), but shorter than the extinct araripemydid *Araripemys barretoi* (67%).

Initial studies of suction feeding in extant chelids placed much emphasis on their ability to feed upon fish [93]. Subsequent analysis of stomach contents from chelids caught in the wild have since confirmed a specialized piscivorous diet for *Chelus fimbriatus* [94]. Studies of *Chelodina* spp. and *Hydromedusa* spp., however, have established a much wider carnivorous dietary spectrum that includes small aquatic invertebrates (mostly insect larvae) and vertebrates [95–98]. A similarly broad, opportunistic food palate seems plausible for *Sahonachelys mailakavava* as well.

Although a comprehensive functional morphological analysis of feeding strategies across all lineages is still outstanding, fossil and recent pan-pelomedusoids appear to mostly employ a ram-feeding strategy. Many taxa show dramatic adaptations, in the form of flat, expanded triturating surfaces, for crushing hard-shelled prey items [5]. In addition to *Sahonachelys mailakavava*, the only other exceptions to a ram-feeding ecology within pan-pelomedusoids are the araripemydids *Araripemys barretoi* and *Taquetochelys decorata* [5,92]. Interestingly, outside of *Pan-Pelomedusoides*, specialized suction feeding is prevalent among the basal-branching groups of chelids and stem pleurodires [93,99]. Although the possibility remains that specialized suction feeding was plesiomorphically retained in *Sahonachelys mailakavava*, numerous structural differences (varying neck lengths, cranial proportions and neck motion) coupled with the prevalence of non-specialized morphologies among other early pelomedusoids make it more plausible that specialized suction feeding evolved multiple times across the pleurodiran tree. Although this suggests great evolutionary dynamism, we find it notable that *Sahonachelys mailakavava* is the only known crown pelomedusoid to have evolved this feeding strategy. This further highlights the uniqueness of taxa in the vertebrate faunal assemblage recovered from the Late Cretaceous of Madagascar (e.g. [25,26,31,32,100–102]).

# 6. Conclusion

We here describe a new species of pelomedusoid turtle, *Sahonachelys mailakavava*, based on a near-complete skeleton from the Maevarano Formation of northwestern Madagascar, which has been dated Late Cretaceous (Maastrichtian). The new species can easily be diagnosed relative to all other named pleurodires as it shows a large number of unique characters in both the shell and cranium. A phylogenetic analysis using weighted parsimony indicates that *Sahonachelys mailakavava* is sister to *Sokatra antitra* and that these coeval Madagascan turtles are the only known representatives of the newly recognized clade *Sahonachelyidae*. This clade is sister to the group formed by *Bothremydidae* and *Podocnemidoidae* within crown *Pelomedusoides*. The most conspicuous characteristic of this clade is the presence of an extended posterior process that is formed by the maxilla and protrudes deeply into the lower temporal emargination. A number of highly unusual morphological features suggest that *Sahonachelys mailakavava* was a specialized, aquatic, suction-feeding species that fed upon moving prey. The specialized feeding strategy further highlights the uniqueness of Late Cretaceous Madagascan faunas, as no other crown pelomedusoid is known to have developed this method of prey capture.

Ethics. Our work involved no live animals and we were not required to complete an ethical assessment prior to conducting our research.

Data accessibility. The µCT slice data, scanning parameters and three-dimensional models generated from the slice data are available for download at MorphoSource: https://www.morphosource.org/Detail/MediaDetail/Show/media_id/88408.

The data are provided in electronic supplementary material [103].

Authors' contributions. D.W.K. and L.J.R. organized field programme. D.W.K., T.R.L. and W.G.J. conceived project. W.G.J., D.W.K. and Y.R. obtained µCT scans. Y.R. and S.W.E. segmented µCT data and produced three-dimensional models and digital renderings. W.G.J. and T.R.L. obtained images of specimen and assembled figures. W.G.J., T.R.L., Y.R. and S.W.E. scored phylogenetic matrix. W.G.J., D.W.K. and S.W.E. wrote primary draft of manuscript. All authors edited and approved final version.

Competing interests. We declare we have no competing interests.

Funding. D.W.K. by grants from the U.S. National Science Foundation (EAR-1528273 and EAR-1664432). W.G.J., S.W.E. and Y.R. were supported by a grant from the Swiss National Science Foundation (SNF 200021_178780/1).

Acknowledgements. We thank Joseph Groenke (Ohio University), Natalie Toth (DMNS), Salvador Bastien (DMNS) and Barbara Pittman (DMNS) for mechanical preparation of the specimen described herein. Raymond Rogers (Macalester College) is thanked for discussion concerning depositional environment. We are grateful to Lucille Betti-Nash for drafting figure 1a, Michael D'Emic (Adelphi University) for the photograph in figure 1b, and Patrick O'Connor (Ohio University) for the photograph in figure 1c. Rick Wicker (DMNS) is thanked for the amazing photographs of the specimen used in figures 3, 9 and 10. Lindsay Dougan (Digital Research Laboratory, DMNS) is acknowledged for initial segmentation and Matthew Colbert (University of Texas High-Resolution X-ray Computed Tomography Facility) for scanning. The mesmerizing artwork provided in figure 11 was created by Andrey Atuchin. We finally thank all individuals of the 2015 Mahajanga Basin field crew involved in the collecting of this specimen. Adán Pérez-García, Pedro Romano and an anonymous reviewer are thanked for insightful comments that helped improve the quality of this manuscript.

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
