## [Peer Review File · Royal Society Open Science]

Review History

RSOS-210098.R0 (Original submission)

Review form: Reviewer 1 (Michael Benton)

Is the manuscript scientifically sound in its present form?

Yes

Are the interpretations and conclusions justified by the results?

Yes

Is the language acceptable?

Yes

Do you have any ethical concerns with this paper?

Yes

Have you any concerns about statistical analyses in this paper?

No

Recommendation?

Accept with minor revision (please list in comments)

Comments to the Author(s)

The paper presents a new species of fossil turtle based on excellent fossil material from the Cretaceous of Madagascar. The paper includes a full description of the material, which is professional and carefully documented, and the analytical part includes a phylogenetic analysis, which again is professional and excellent, and this leads to consequences on evolution of feeding modes among pleurodiran turtles. The author team includes world-leading experts on fossil turtles, turtle phylogeny, the Madagascar Cretaceous faunas, and interpretation of these fossils. The authors establish a new family, a new genus, and a new species.

Suggested improvements

Pages 4, 5: iron out repetitions about Maevarano Formation (Maastrichtian age; faunal list) between Introduction and Geological setting.

Page 5: Geological setting [not settings]

Page 10, line 6: Sahonachelyidae [not italic]; also check wrongly italicized higher taxon names at lines 26, 58, page 11, lines 3, 14, 15, 17, 28, 29, etc... page 47, line 3] or is it now appropriate to italicize these?]

The figures are lovely, especially the final, reconstruction, figure.

Review form: Reviewer 2

Is the manuscript scientifically sound in its present form?

Yes

Are the interpretations and conclusions justified by the results?

Yes

Is the language acceptable?

Yes

Do you have any ethical concerns with this paper?

No

Have you any concerns about statistical analyses in this paper?

No

Recommendation?

Accept with minor revision (please list in comments)

Comments to the Author(s)

In my opinion, this is an interesting and well-written paper. Therefore, I recommend its publication in this journal.

I indicate here some minor suggestions:

- I recommend the authors to add a diagnosis for the new clade Sahonachelyidae. Some or all of the characters that could diagnose this clade are indicated throughout the Discussion section, but they should be grouped, as a diagnosis, in the Systematic paleontology section.
- The order of the figures is not the same as that in which they are cited: Figure 10 is one of the first to be referred.
- I recommend including all the comparisons between the new species and its sister taxon Sokatra in the Discussion section but not in the Description section (and, to avoid repetitions, remove all of them from the Description section).
- I suggest removing the reference to "Bronzati et al. in review": Unaccepted or unpublished papers should not be referred.
- A minor mistake is detected in Figure 8: neural VI is indicated as IV. A recommendation in that figure: the abbreviations of all the scutes are written with only the initial letter in capital letters, except those for GU and EG. Could this be modified for those two scutes?

I'm available for any questions that the editor or authors may have in relation to my review.

Review form: Reviewer 3 (Pedro Romano)

Is the manuscript scientifically sound in its present form?

Yes

Are the interpretations and conclusions justified by the results?

Yes

Is the language acceptable?

Yes

Do you have any ethical concerns with this paper?

No

Have you any concerns about statistical analyses in this paper?

Yes

Recommendation?

Accept with minor revision (please list in comments)

Comments to the Author(s)

I made comments directly on the manuscript (see Appendix A). They are highlighted with yellow text marker and comments. the most relevant suggestions/comments are:

1) I think it is better to suggest a new species name included in the already established genus Sokatra. The decision of creating a new genus and species is arbitrary and there is no direct evidence for it. Sokatra is a monotypic genus and including a second species found as sister to the type species (Sokatra antitra) is a better taxonomic decision in my opinion. See further comments along the text.

2) Neck measurements M&M and results need to be reviewed.

2.1) Imprecision: The most inaccurate/imprecise would be measuring with ImageJ and caliper and mixing different measurements from different measuring methods. This is not the case and the error is minimized. See Mariani & Romano, 2017 DOI 10.7717/peerj.2890. All measurements were performed by the same author? Better, to avoid different individual error input. I recommend to take the same measurements more than one time (x3?) and use mean values as entries. An ANOVA can be done to detect differences and measure the error. The statement that this measurements "should be seen as approximations" is confusing. Should I trust the results? If you present a description of the measurements "errors" (basic statistics, including standard deviation) I can check if the data is ok or not.

2.2) Contradiction: In M&M it is stated that "missing neck vertebrae was estimated based on averages of the preserved vertebrae" (see page 9 of the manuscript). In results (see page 46 of the manuscript) it is stated that it was assumed (for the new species) "... the remaining [i.e., excluding atlas and axis] six cervicals had similar dimensions, we calculated a neck length of approximately 14cm." Thus, the missing entries were estimated differently in all species used for comparisons? Therefore, different assumptions were made and this is reckless. if I understand this correctly, assuming C3-C8 equal midline length of C2 (axis) means that C2-C8 has 126mm (18mm x7). Thus, C1 (atlas) has 14mm (the measure was not informed by the authors). If we use the "averages of the preserved vertebrae" C3-C8 should be estimated as 16mm each, leading to a neck of ~128mm and a neck to carapace ratio ~50%. This is closer to *Chelonoidis denticulata* (48%), *Pelomedusa subrufa* (49%), *Cuora amboinensis* (51%), and *Geoclemys hamiltonii* (51%). Besides that, C3 is partially preserved and according to figure 9b, I would say it is possibly shorter than axis (C2). The atlas (C1) is clearly shorter than 14mm. Ultimately, the morphometrics need to be revised and these estimates for UA 10581 are not reliable.

2.3) Persuasion: In the same paragraph at page 46, the authors say "... gives a neck to carapace ratio of about 55%, which is comparable to the ratio of long necked chelids, particularly *Chelys fimbriatus* (59%...". However, it is even closer to *Terrapene carolina* (53%), *Kinixys belliana* (54%), *Sacalia quadriocellata* (55%), *Sternotherus depressus* (56%), and *Kinosternon baurii* (57%). The direct comparison to *matamata* is misleading to reader and the authors are ignoring their own data to inform a precise comparison. This seems to be skewed to provide stronger support to their conclusions of parallel evolution. It is not necessary because the qualitative description and explanation about putative adaptation to suction feeding in this new species is well written and funded in the literature.

3) Phylogeny.

3.1) I was able to replicate this search using TNT v1.5. However, some small details could be provided. For instance, I found 29 MPTs of 55.60618. I used random seed as 1 and saving only 10 trees per replica; therefore, using default setting given that these configurations were not mentioned in M&M. I found the same topology strict consensus (provided as suppl material). After increasing the search strategies settings, by using random seed zero, saving up to 200 trees in each replica and collapsing trees after search, I found 33 MPTs. It is important to notice that the general memory should be adjusted to more than 100 trees (TNT default) in order to achieve these results. I've also run an analysis assuming a priori equal weights (=1) to all characters. This resulted on 5400 MPTs of 1295 steps. The topology is different (specially not recovering Chelidae as a clade), but there are several similarities to tree found in the original analysis. It is important

to notice that the new species is found as sister to Sokatra antitra and that this clade is nested in a polytomy with Atolchelys lepida + Podocnemidoidea+Bothremydidae clade. Thus, the assumption about weighting of characters do not change the general structure of the preferred phylogeny (fig 10).

3.2) Caninemys tridentata is missing in the full strict consensus topology provided as supplementary material. BTW, not following Cadena, who considered Caninemys tridentata as junior synonym of Stupendemys geographicus?

3.3) There is an "alien" topology in the supplement material provided: TNT file with the character/taxon matrix. Please, check it. This is in the "tread block" at the end of the file, right after the list of multistate characters that should be treated as additive (ccode block). I'm attaching this topology, that is the following (parenthetical):

```
(((((23 (22 21)) (76 (77 (72 (69 (73 71 70 (75 74))))))) (((3 (6 (4 5))) 12) ((16 15) (((14 13) (8 7)) (11 (10 9)))))) 1) ((((((25 24) (27 26)) (17 (20 19))) 2) 78) ((28 18) (33 ((30 31 (37 ((34 (36 35)) (((40 38 39) (43 (42 (41 ((49 48) ((47 46) (45 44))))))) ((60 59) (53 (54 (55 ((58 (56 57)) (52 (51 50))))))))))))) (61 (62 (63 (65 (64 ((66 (67 68)) (79 ((80 81 83 82) (86 89 84 87 88 85 (90 (95 (96 (98 ((91 (92 (103 93 94))) (102 100 101 (99 97))))))))))))))))))
```

That being said, it is important to highlight the quality of the manuscript. The description of the new species is presented in a very detailed and careful way and very well illustrated. The hypothesis of convergent evolution of suction feeding is supported by clear evidence and convincing. All abovementioned comments are minor details that I hope can improve even more the quality of the manuscript. Congratulations to the authors for this contribution.

No need to remain anonymous.
best regards,
Pedro Romano

Decision letter (RSOS-210098.R0)

Dear Dr Joyce

On behalf of the Editors, we are pleased to inform you that your Manuscript RSOS-210098 "A new pelomedusoid turtle from the Late Cretaceous of Madagascar provides evidence for convergent evolution of suction feeding among pleurodires" has been accepted for publication in Royal Society Open Science subject to minor revision in accordance with the referees' reports. Please find the referees' comments along with any feedback from the Editors below my signature.

Please submit your revised manuscript and required files (see below) no later than 7 days from today's (ie 07-Apr-2021) date. Note: the ScholarOne system will 'lock' if submission of the

revision is attempted 7 or more days after the deadline. If you do not think you will be able to meet this deadline please contact the editorial office immediately.

on behalf of Professor Kevin Padian (Subject Editor)
openscience@royalsociety.org

Editor comments to Author:

The reviewers are very pleased with your submission, and so we are glad to accept it. However they bring up a number of relatively minor comments that we would ask you to address in your revisions, and please include a memo explaining how you have dealt with them. Congratulations and best wishes.

Reviewer comments to Author:

Reviewer: 1
Comments to the Author(s)

The paper presents a new species of fossil turtle based on excellent fossil material from the Cretaceous of Madagascar. The paper includes a full description of the material, which is professional and carefully documented, and the analytical part includes a phylogenetic analysis, which again is professional and excellent, and this leads to consequences on evolution of feeding modes among pleurodiran turtles. The author team includes world-leading experts on fossil turtles, turtle phylogeny, the Madagascar Cretaceous faunas, and interpretation of these fossils. The authors establish a new family, a new genus, and a new species.

Suggested improvements

Pages 4, 5: iron out repetitions about Maevarano Formation (Maastrichtian age; faunal list) between Introduction and Geological setting.

Page 5: Geological setting [not settings]

Page 10, line 6: Sahnachelyidae [not italic]; also check wrongly italicized higher taxon names at lines 26, 58, page 11, lines 3, 14, 15, 17, 28, 29, etc... page 47, line 3] or is it now appropriate to italicize these?]

The figures are lovely, especially the final, reconstruction, figure.

Reviewer: 2

Comments to the Author(s)

In my opinion, this is an interesting and well-written paper. Therefore, I recommend its publication in this journal.

I indicate here some minor suggestions:

- I recommend the authors to add a diagnosis for the new clade Sahnachelyidae. Some or all of the characters that could diagnose this clade are indicated throughout the Discussion section, but they should be grouped, as a diagnosis, in the Systematic paleontology section.
- The order of the figures is not the same as that in which they are cited: Figure 10 is one of the first to be referred.
- I recommend including all the comparisons between the new species and its sister taxon Sokatra in the Discussion section but not in the Description section (and, to avoid repetitions, remove all of them from the Description section).
- I suggest removing the reference to "Bronzati et al. in review": Unaccepted or unpublished papers should not be referred.
- A minor mistake is detected in Figure 8: neural VI is indicated as IV. A recommendation in that figure: the abbreviations of all the scutes are written with only the initial letter in capital letters, except those for GU and EG. Could this be modified for those two scutes?

I'm available for any questions that the editor or authors may have in relation to my review.

Reviewer: 3

Comments to the Author(s)

I made comments directly on the manuscript. They are highlighted with yellow text marker and comments. the most relevant suggestions/comments are:

1) I think it is better to suggest a new species name included in the already established genus Sokatra. The decision of creating a new genus and species is arbitrary and there is no direct evidence for it. Sokatra is a monotypic genus and including a second species found as sister to the type species (Sokatra antitra) is a better taxonomic decision in my opinion. See further comments along the text.

2) Neck measurements M&M and results need to be reviewed.

2.1) Imprecision: The most inaccurate/imprecise would be measuring with ImageJ and caliper and mixing different measurements from different measuring methods. This is not the case and the error is minimized. See Mariani & Romano, 2017 DOI 10.7717/peerj.2890. All measurements were performed by the same author? Better, to avoid different individual error input. I recommend to take the same measurements more than one time (x3?) and use mean values as entries. An ANOVA can be done to detect differences and measure the error. The statement that

this measurements "should be seen as approximations" is confusing. Should I trust the results? If you present a description of the measurements "errors" (basic statistics, including standard deviation) I can check if the data is ok or not.

2.2) Contradiction: In M&M it is stated that "missing neck vertebrae was estimated based on averages of the preserved vertebrae" (see page 9 of the manuscript). In results (see page 46 of the manuscript) it is stated that it was assumed (for the new species) "... the remaining [i.e., excluding atlas and axis] six cervicals had similar dimensions, we calculated a neck length of approximately 14cm." Thus, the missing entries were estimated differently in all species used for comparisons? Therefore, different assumptions were made and this is reckless. If I understand this correctly, assuming C3-C8 equal midline length of C2 (axis) means that C2-C8 has 126mm (18mm x7). Thus, C1 (atlas) has 14mm (the measure was not informed by the authors). If we use the "averages of the preserved vertebrae" C3-C8 should be estimated as 16mm each, leading to a neck of ~128mm and a neck to carapace ratio ~50%. This is closer to *Chelonoidis denticulata* (48%), *Pelomedusa subrufa* (49%), *Cuora amboinensis* (51%), and *Geoclemys hamiltonii* (51%). Besides that, C3 is partially preserved and according to figure 9b, I would say it is possibly shorter than axis (C2). The atlas (C1) is clearly shorter than 14mm. Ultimately, the morphometrics need to be revised and these estimates for UA 10581 are not reliable.

2.3) Persuasion: In the same paragraph at page 46, the authors say "... gives a neck to carapace ratio of about 55%, which is comparable to the ratio of long necked chelids, particularly *Chelys fimbriatus* (59%...". However, it is even closer to *Terrapene carolina* (53%), *Kinixys belliana* (54%), *Sacalia quadriocellata* (55%), *Sternotherus depressus* (56%), and *Kinosternon baurii* (57%). The direct comparison to *matamata* is misleading to reader and the authors are ignoring their own data to inform a precise comparison. This seems to be skewed to provide stronger support to their conclusions of parallel evolution. It is not necessary because the qualitative description and explanation about putative adaptation to suction feeding in this new species is well written and funded in the literature.

3) Phylogeny.

3.1) I was able to replicate this search using TNT v1.5. However, some small details could be provided. For instance, I found 29 MPTs of 55.60618. I used random seed as 1 and saving only 10 trees per replica; therefore, using default setting given that these configurations were not mentioned in M&M. I found the same topology strict consensus (provided as suppl material). After increasing the search strategies settings, by using random seed zero, saving up to 200 trees in each replica and colapsing trees after search, I found 33 MPTs. It is important to notice that the general memory should be adjusted to more than 100 trees (TNT default) in order to achieve these results. I've also run an analysis assuming a priori equal weights (=1) to all characters. This resulted on 5400 MPTs of 1295 steps. The topology is different (specially not recovering Chelidae as a clade), but there are several similarities to tree found in the original analysis. It is important to notice that the new species is found as sister to *Sokatra antitra* and that this clade is nested in a polytomy with *Atolchelys lepida* + *Podocnemidoidea*+*Bothremyidae* clade. Thus, the assumption about weighting of characters do not change the general structure of the preferred phylogeny (fig 10).

3.2) *Caninemys tridentata* is missing in the full strict consensus topology provided as supplementary material. BTW, not following Cadena, who considered *Caninemys tridentata* as junior synonym of *Stupendemys geographicus*?

3.3) There is an "alien" topology in the supplement material provided: TNT file with the character/taxon matrix. Please, check it. This is in the "tread block" at the end of the file, right

after the list of multistate characters that should be treated as additive (ccode block). I'm attaching this topology, that is the following (parenthetical):

(((((23 (22 21)) (76 (77 (72 (69 (73 71 70 (75 74))))))) ((3 (6 (4 5))) 12) ((16 15) (((14 13) (8 7)) (11 (10 9)))))) 1) ((((((25 24) (27 26)) (17 (20 19))) 2) 78) ((28 18) (33 ((30 31 (37 ((34 (36 35)) (((40 38 39) (43 (42 (41 ((49 48) ((47 46) (45 44))))))) ((60 59) (53 (54 (55 ((58 (56 57)) (52 (51 50)))))))))) (61 (62 (63 (65 (64 ((66 (67 68)) (79 ((80 81 83 82) (86 89 84 87 88 85 (90 (95 (96 (98 (91 (92 (103 93 94))) (102 100 101 (99 97))))))))))))))))))

That being said, it is important to highlight the quality of the manuscript. The description of the new species is presented in a very detailed and careful way and very well illustrated. The hypothesis of convergent evolution of suction feeding is supported by clear evidence and convincing. All abovementioned comments are minor details that I hope can improve even more the quality of the manuscript. Congratulations to the authors for this contribution.

No need to remain anonymous.

best regards,

Pedro Romano

===PREPARING YOUR MANUSCRIPT===

===PREPARING YOUR REVISION IN SCHOLARONE===

To revise your manuscript, log into <https://mc.manuscriptcentral.com/rsos> and enter your Author Centre - this may be accessed by clicking on "Author" in the dark toolbar at the top of the

page (just below the journal name). You will find your manuscript listed under "Manuscripts with Decisions". Under "Actions", click on "Create a Revision".

<https://royalsociety.org/journals/authors/author-guidelines/#supplementary-material> to include a suitable title and informative caption. An example of appropriate titling and captioning may be found at https://figshare.com/articles/Table_S2_from_Is_there_a_trade-off_between_peak_performance_and_performance_breadth_across_temperatures_for_aerobic_sc_ope_in_teleost_fishes_/3843624.

Author's Response to Decision Letter for (RSOS-210098.R0)

See Appendix B.

Decision letter (RSOS-210098.R1)

Dear Dr Joyce,

I am pleased to inform you that your manuscript entitled "A new pelomedusoid turtle from the Late Cretaceous of Madagascar provides evidence for convergent evolution of suction feeding among pleurodires" is now accepted for publication in Royal Society Open Science.

on behalf of Kevin Padian (Subject Editor)
openscience@royalsociety.org

Appendix A**ROYAL SOCIETY
OPEN SCIENCE****A new pelomedusoid turtle from the Late Cretaceous of
Madagascar provides evidence for convergent evolution of
suction feeding among pleurodires**

Journal:	Royal Society Open Science
Manuscript ID	RSOS-210098
Article Type:	Research
Date Submitted by the Author:	21-Jan-2021
Complete List of Authors:	Joyce, Walter; Universität Freiburg ROLLOT, Yann; Universität Freiburg, Departement für Geowissenschaften Evers, Serjoscha W.; Universität Freiburg, Departement für Geowissenschaften Lyson, Tyler; Denver Museum of Nature & Science Rahantarisoa, Lydia J.; Université d'Antananarivo, Département de Sciences de la Terre et de l'Environnement Krause, David W.; Denver Museum of Nature & Science, Department of Earth Sciences; Stony Brook University, Department of Earth Sciences
Subject:	evolution < BIOLOGY, taxonomy and systematics < BIOLOGY, palaeontology < BIOLOGY
Keywords:	Late Cretaceous, Maastrichtian, Maevarano Formation, Testudines, Pleurodira, Sahnachelyidae
Subject Category:	Organismal and Evolutionary Biology

Author-supplied statements

Relevant information will appear here if provided.

Ethics

Does your article include research that required ethical approval or permits?:

This article does not present research with ethical considerations

Statement (if applicable):

CUST_IF_YES_ETHICS :No data available.

Data

It is a condition of publication that data, code and materials supporting your paper are made publicly available. Does your paper present new data?:

Yes

Statement (if applicable):

The μ CT slice data, scanning parameters, and 3D models generated from the slice data are available for download at MorphoSource:

https://www.morphosource.org/Detail/MediaDetail/Show/media_id/88408.

Conflict of interest

I/We declare we have no competing interests

Statement (if applicable):

CUST_STATE_CONFLICT :No data available.

Authors' contributions

This paper has multiple authors and our individual contributions were as below

Statement (if applicable):

D.W.K. and L.J.R. organized field program. D.W.K., T.R.L. and W.G.J. conceived project. W.G.J., D.W.K., and Y.R. obtained μ CT scans. Y.R. and S.W.E. segmented μ CT data and produced 3D models and digital renderings. W.G.J. and T.R.L. obtained images of specimen and assembled figures. W.G.J., T.R.L., Y.R. and S.W.E. scored phylogenetic matrix. W.G.J., D.W.K., and S.W.E. wrote primary draft of manuscript. All authors edited and approved final version.

**A new pelomedusoid turtle, *Sahonachelys mailakavava*, from the Late**
**Cretaceous of Madagascar provides evidence for convergent evolution of**
**specialized suction feeding among pleurodires**

Walter G. Joyce¹, Yann Rollot¹, Serjoscha W. Evers¹, Tyler R. Lyson², Lydia J.
Rahantarisoa³, and David W. Krause^{2,4}

¹Departement für Geowissenschaften, Universität Freiburg, Fribourg, Switzerland

²Department of Earth Sciences, Denver Museum of Nature & Science, Denver, CO, USA

³Département de Sciences de la Terre et de l'Environnement, Université d'Antananarivo,
Antananarivo, Madagascar

⁴Department of Anatomical Sciences, Stony Brook University, Stony Brook, NY, USA

Keywords: Late Cretaceous, Maastrichtian, Maevarano Formation, *Testudines*, *Pleurodira*,
*Sahonachelyidae*

**ABSTRACT**

The Maevarano Formation in northwestern Madagascar has yielded a series of exceptional
fossils over the course of the last three decades that provide important insights into the
evolution of insular ecosystems during the latest Cretaceous (Maastrichtian). We here
describe a new genus and species of pelomedusoid turtle from this formation, *Sahonachelys*
*mailakavava*, based on a nearly complete skeleton. A phylogenetic analysis suggests close
affinities of *Sahonachelys mailakavava* with the coeval Madagascan *Sokatra antitra*. These
two taxa are the only known representatives of the newly recognized clade *Sahonachelyidae*,
which is sister to the speciose clade formed by *Bothremydidae* and *Podocnemidoidae*. A

close relationship with coeval Indian turtles of the clade *Kurmademydini* is notably absent. A
functional assessment suggests that *Sahonachelys mailakavava* was a specialized suction
feeder that preyed upon small-bodied invertebrates and vertebrates. This is a unique feeding
strategy among crown pelomedusoids that is convergent upon that documented in numerous
other clades of turtles and that highlights the distinct evolutionary pathways taken by
Madagascan vertebrates.

**1. Introduction**

Extant pelomedusoid turtles are found across much of Africa, Madagascar, and South
America. Although the clade diverged as early as the Early Cretaceous, approximately the
same time as chelids, trionychids, and durocryptodires (Joyce et al. 2013a; Pereira et al.
2017; Ferreira et al. 2018a), today they comprise only about 10 percent of species diversity
and 5 percent of generic diversity (TTWG, 2017). Over the course of the last 25 years,
however, an astounding diversity of fossil forms has been documented from Cretaceous to
Paleocene strata (e.g., Gaffney et al. 2006, 2011; Romano et al. 2014; Ferreira et al. 2015,
2018c, 2018b; Bourque 2016; Joyce et al. 2016a; Pérez-García 2016, 2017, 2019, online;
Pérez-García et al., 2017; Cadena et al. 2020; Lapparent de Broin et al. 2020; Joyce and
Bandyopadhyay, 2020; López-Conde et al. 2021). These fossils not only expand the range of
pelomedusoids to Arabia, the Caribbean, Europe, India, and North America, but also
document an incredible array of diversity and disparity within the group. Although a formal
ecomorphological analysis is still outstanding for the entire clade, fossil and extant
pelomedusoids generally possess high skulls with well-developed triturating surfaces that
suggest herbivorous, omnivorous, and durophagous diets (Ferreira et al. 2015; Foth et al.
2017).

The Maevarano Formation in northwestern Madagascar, which is dated Late Cretaceous
(Maastrichtian) has yielded a rich continental fauna consisting of fishes, frogs, lepidosaurs,
archosaurs, and mammals (Krause et al. 2020a). The formation is particularly well known for
yielding exquisitely preserved small-bodied vertebrates, including the frog *Beelzebubo*
*ampinga* (Evans et al. 2014), the cordylid lizard *Konkasaurus mahalana* (Krause et al.,
2003), the notosuchian *Araripesuchus tsangatsangana* (Turner 2006), the paravian theropod
*Rahonavis ostromi* (Forster et al., 1998, 2020), the enantiornithine theropod *Falcatakely*
*forsterae* (O'Connor et al. 2020), and the gondwanatherian mammals *Vintana sertichi*
(Krause et al., 2014) and *Adalatherium hui* (Krause et al., 2020a). In addition to fragmentary
material that may represent additional species (Krause et al., 2020a), the Maevarano
Formation has also yielded cranial material of the bothremydid turtle *Kinkonychelys rogersi*
(Gaffney et al. 2009), the basal pelomedusoid *Sokratra antitra* (Gaffney and Krause, 2011),
and an unnamed podocnemidoid (Gaffney and Foster, 2003).

In 2015, a pelomedusoid turtle skeleton was discovered while removing overburden from
a productive archosaur site within the Maevarano Formation of northwestern Madagascar.
The specimen is not only unusual for its fragility and completeness, but also in displaying
numerous morphological adaptations consistent with specialized suction feeding. The
purpose of this contribution is to describe this specimen as a new genus and species of turtle,
*Sahonachelys mailakavava*, to investigate its phylogenetic relationships, and assess its
paleoecology.

2. Geological settings

The holotype and only known specimen of *Sahonachelys mailakavava* gen. et sp. nov.,
Université d'Antananarivo (UA) 10581, was discovered in June, 2015 at locality MAD05-38
in the Berivotra Study Area of the Mahajanga Basin, northwestern Madagascar, while

removing overburden less than 1 m above a layer rich in dinosaur and crocodyliform fossils
(figure 1). Locality MAD05-38 occurs in the Anembalemba Member of the Maevarano
(figure 1). Locality MAD05-38 occurs in the Anembalemba Member of the Maevarano
Formation (Rogers et al., 2000). **Locality coordinates** are on file at the Denver Museum of
Nature & Science and the Université d'Antananarivo and are available to qualified
investigators.

The Maevarano Formation is of Maastrichtian (latest Cretaceous) age (see summary of
geochronological data in Krause et al., 2020b). The vertebrate assemblage from the formation
includes dipnoan fishes (1 sp.), actinopterygian fishes (10 spp.), frogs (2 spp.), turtles (**5 spp.,**
**including *S. mailakavava***), snakes (6 spp.), non-ophidian squamates (1 sp.), crocodyliforms
(6 spp.), titanosaurian sauropod dinosaurs (2 spp.), non-avian theropod dinosaurs (3 spp.),
avialans (6 spp.), and mammals (7 spp.), for a total of 51 currently known species. A recently
revised faunal list is presented in Krause et al. (2020a: table S2). **Over 100 fossil vertebrate**
**specimens have been recovered to date from locality MAD05-38.** The fauna preliminarily
identified and associated with *S. mailakavava* at the site includes the crocodyliform
*Miadanasuchus oblita*, the titanosaurian sauropod *Rapetosaurus krausei*, and the non-avian
abelisauroid theropods *Majungasaurus crenatissimus* and *Masiakasaurus knopfleri*, in
addition to possible paravian material.

The Maevarano Formation was deposited at a time when northwestern Madagascar had a
highly seasonal (pronounced dry and wet seasons), semi-arid climate and was positioned at
approximately 30°S latitude (15° farther south than today), in the subtropical desert belt (see
summaries in Krause et al., 2020a, 2020b). Madagascar was isolated in the Indian Ocean,
having been separated from the African mainland before 165 Myr ago, from Antarctica-
Australia at ~124 Myr ago, and the Indian subcontinent at ~88 Myr ago, approximately 20
million years earlier than when the Maevarano Formation was deposited (Krause et al.,
2020a).

**The Anembalemba Member of the Maevarano Formation in the Berivotra Study Area**
**ranges from 10 to 15 m in thickness. The member is primarily composed of two sandstone**
**facies, Facies 1 and Facies 2.** Facies 1 consists of light gray to white, fine- to coarse-grained,
poorly sorted sandstone with small- to medium-scale tabular and trough cross stratification
and is thought to represent normal flow in an aggrading stream channel. Facies 2, on the
other hand, consists of olive green, fine- to coarse-grained, clay-rich sandstone, which has
been interpreted as representing a debris flow setting (Rogers et al., 2000; Rogers, 2005). To
date, all articulated or associated vertebrate fossils, in addition to isolated specimens, in the
Anembalemba Member have been recovered from Facies 2, while only isolated specimens
have been recovered from Facies 1. **UA 10581** was collected from a somewhat less massive
variant of Facies 2.

**3. Materials and methods**

**3.1. Institutional abbreviation**

UA, Université d'Antananarivo, Antananarivo, Madagascar.

**3.2. Nomenclature**

We here follow the anatomical terminology of Gaffney (1979) for cranial structure, with
modifications from Rabi et al. (2013) and Rollot et al. (2021) in regards to the carotid and
facial nerve canal systems. We here follow the taxonomic nomenclature of Joyce et al.
(2021). All names are therefore phylogenetically defined clade names highlighted through the
use of italics, with exception of *Araripemydidae* for which we utilized the taxonomic concept
of Meylan (1996) and *Podocnemidoidae*, for which we utilize the taxonomic concept of
Ferreira et al. (2018a).

3.3. μ CT scanning and 3D modeling

High-resolution X-ray micro-computed tomography (μ CT) scans were obtained for the cranium and hyoids of **UA 10581** at the University of Texas High-Resolution X-ray Computed Tomography Facility using an NSI scanner with a 225-kV Feinfocus microfocal source. The specimen was scanned using a beam energy of 160 kV, a current of 0.2 mA, and an aluminum filter, resulting in an isometric voxel size of 0.0285 mm. The resulting slice images were segmented using the software Amira (<http://www.amira.com>) and 3D models were exported in .ply format. The digital renderings utilized in the figures were compiled using the software Blender v. 2.71 (<http://www.blender.org/>). μ CT-slice data as well as the 3D models are deposited at MorphoSource

(https://www.morphosource.org/Detail/MediaDetail/Show/media_id/88408).

3.4. Phylogenetic analysis

We explored the phylogenetic relationships of *Sahonachelys mailakavava* gen. et sp. nov. by scoring its morphology within the pleurodire matrix of Joyce and Bandyopadhyay (2020), which is based on the work of Bona and de la Fuente (2005), Gaffney et al. (2006, 2011), and Ferreira et al. (2018a). This matrix was chosen because it includes the most recent scoring for all known fossil pelomedusoids from the Late Cretaceous of Madagascar and India, in particular *Jainemys pisdurensis*, *Kinkonychelys rogersi*, *Kurmademys kallamedensis*, *Sankuchemys sethnai*, and *Sokratra antitra*. The matrix was expanded to include 21 additional characters based on μ CT scans that had recently been developed by Hermanson et al. (2020). **Finally, a new character pertaining to the presence of an elongate posterior process of the maxilla was added to highlight similarities between *Sahonachelys mailakavava* and *Sokratra antitra*.** The final matrix is provided in the electronic supplementary material and includes the full list of characters and character states.

The matrix was subjected to a parsimony analysis using the traditional search options of
the software TNT (Goloboff et al., 2008). All characters that form morphoclines were
ordered and are noted as such in the matrix. *Proganochelys quenstedti* was selected as the
outgroup. Light implied weighting was implemented with a k value of 12 (Goloboff et al.,
2018), and 1,000 replicates of random addition sequences were followed by a round of tree
bisection and reconnection.

19 3.5. Measure of relative neck length

We collected a data set pertaining to the relative length of the neck of a sample of fossil
and recent turtles. For this purpose, we took measurement from pictures using ImageJ
(<https://imagej.nih.gov/ij/>) for the cumulative length of all cervical centra and the midline
length of the carapace. Given the imprecision associated with this methodology, all
measurements should be seen as approximations. For fossil turtles, the length of missing neck
vertebrae was estimated based on averages of the preserved vertebrae. The final data set
[revised manuscript text omitted]

*rogersi* (Gaffney et al. 2009) and the unnamed podocnemidoid (Gaffney and Foster, 2003)
from the Maevarano Formation.

**The immediate sister group relationship of *Sahonachelys mailakavava* gen. et sp. nov.**
**with *Sokratra antitra* is supported by a reduced contribution of the maxilla to the floor of the**
**orbit (homoplastically developed in some large-bodied taphrosphyines), and, most apparent,**
**the presence of distinct posterior process of the maxilla.** These two turtles are the only known
representatives of the clade *Sahonachelyidae*. *Sahonachelys mailakavava* differs from
*Sokratra antitra* by having a much flatter and broader cranium with more dorsally oriented
orbits, the presence of a prefrontal-palatine contact, the absence of parietal-palatine contact,
by having narrower triturating surfaces that lack a lingual ridge, a narrower midline contact
of the palatines, the presence of a distinct supramaxillary artery sulcus on the ventral side of
the jugal, and the exposure of the prootic anterior and posterior to the foramen posterius
canalis carotici interni.

The μ CT scan data suggest a number of unusual characteristics in the cranium of
*Sahonachelys mailakavava*, but the lack of comparative data, particularly for *Sokratra antitra*,
makes it impossible to determine if these represent autapomorphies, synapomorphies, or
individual variations. These morphological features include the presence of a small, slit-like
posterior opening to the antrum postoticum between the squamosal and quadrate, a quadrate
contribution to the trigeminal foramen, a quadrate contribution to the lateral semicircular
canal, a foramen in the floor of the sulcus cavernosus that communicates with the facial
and/or carotid canal system, an abducens sulcus on the lateral margin of the rostrum
basisphenoidale, an extremely voluminous recessus scalae tympani that is possibly associated

with the absence of a glossopharyngeal canal through the opisthotic, and the ventral entry of
the supramaxillary artery.

Our phylogenetic hypothesis broadly corresponds with those presented by Gaffney and
Krause (2011) and Ferreira et al. (2018a) by hypothesizing that *Sahonachelyidae* is sister to
the clade formed by *Podocnemidoidea* and *Bothremydoidea*. This is not particularly surprising,
as we utilized a modified version of the matrix assembled by Ferreira et al. (2018a), which in
turn is a variant of the matrix utilized by Gaffney and Krause (2011). *Sahonachelys*
*mailakavava*, therefore, does not appear to be a tree-changing taxon, but rather hints at a
growing stability in the understanding of pelomedusoid relationships.

**We note two interesting implications for this topology.** First, the oldest known
representatives of the clade formed by *Podocnemidoidea* and *Bothremydoidea* are known from
the late Early Cretaceous (e.g., **the Albian *Brasilemys josai* and *Cearachelys placidoi***). This
implies a sahonachelyid ghost lineage that extends from at least the Albian to Campanian, a
time span of nearly 45 million years, a fact already alluded to by Gaffney and Krause (2011)
and consistent with phylogenetic assessments for various other vertebrate taxa from the Late
Cretaceous of Madagascar (e.g., Ali and Krause, 2011; Krause et al., 2014, 2019, 2020a).
60
Second, although sampling of vertebrates from the pre-Maastrichtian Late Cretaceous of both
Madagascar and the Indian subcontinent is still very limited, it is notable that no
sahonachelyids have yet been found on the latter **in the reasonably well-sampled**
Maastrichtian. Given that the distribution of continental pleurodires seems to contain a strong
biogeographic signal (Joyce et al., 2016b), a literal interpretation of the topology obtained
herein suggests that Madagascar split from the rest of Gondwana as early as the late Early
Cretaceous. As Madagascar is usually hypothesized to have been connected with the Indian
subcontinent until the Coniacian, at approximately 88 Myr ago (Storey et al., 1995, 1997;
Melluso et al., 1997, 2001, 2009; Torsvik et al., 1998, 2000; Yatheesh et al., 2013; Reeves,

[revised manuscript text omitted]
2010; Krause et al., 2014, 2020a; O'Connor et al. 2020).

**6. Conclusions**

We here describe a new species of pelomedusoid turtle, *Sahonachelys mailakavava*,

based on a near-complete skeleton from the Maevarano Formation of northwestern
Madagascar, which has been dated Late Cretaceous (Maastrichtian). The new species can
easily be diagnosed relative to all other named pleurodires as it shows a large number of
unique characters in both the shell and cranium. A phylogenetic analysis using weighted
parsimony indicates that *Sahonachelys mailakavava* is sister to *Sokratra antitra* and that these
coeval Madagascan turtles are the only known representatives of the newly recognized clade
*Sahonachelyidae*. This clade is sister to the group formed by *Bothremydidae* and
*Podocnemidoidae* within crown *Pelomedusoides*. The most conspicuous characteristic of this
clade is the presence of an extended posterior process that is formed by the maxilla and
protrudes deeply into the lower temporal emargination. A number of highly unusual
morphological features suggest that *Sahonachelys mailakavava* was a specialized, aquatic,
suction-feeding species that fed upon moving prey. The specialized feeding strategy further
highlights the uniqueness of Late Cretaceous Madagascan faunas, as no other crown
pelomedusoid is known to have developed this method of prey capture.

**Ethics.** Our work involved no live animals and we were not required to complete an ethical
assessment prior to conducting our research.

**Data accessibility.** The μ CT slice data, scanning parameters, and 3D models generated from
the slice data are available for download at MorphoSource:

https://www.morphosource.org/Detail/MediaDetail/Show/media_id/88408.

**Authors' contributions.** D.W.K. and L.J.R. organized field program. D.W.K., T.R.L. and
55 W.G.J. conceived project. W.G.J., D.W.K., and Y.R. obtained μ CT scans. Y.R. and S.W.E.
segmented μ CT data and produced 3D models and digital renderings. W.G.J. and T.R.L.

obtained images of specimen and assembled figures. W.G.J., T.R.L., Y.R. and S.W.E. scored
phylogenetic matrix. W.G.J., D.W.K., and S.W.E. wrote primary draft of manuscript. All
authors edited and approved final version.

**Competing interests.** We declare we have no competing interests.

**Funding.** DWK by grants from the U.S. National Science Foundation (EAR-1528273 and
EAR-1664432). W.G.J., S.W.E., and Y.R. were supported by a grant from the Swiss National
Science Foundation (SNF 200021_178780/1).

**Acknowledgements.** We would like to thank Joseph Groenke (Ohio University), Natalie
Toth (DMNS), Salvador Bastien (DMNS), and Barbara Pittman (DMNS) for mechanical
preparation of the specimen described herein. Raymond Rogers (Macalester College) is
thanked for discussion concerning depositional environment. We are grateful to Lucille Betti-
Nash for drafting figure 1a, Michael D’Emic (Adelphi University) for the photograph in
figure 1b, and Patrick O’Connor (Ohio University) for the photograph in figure 1c. Rick
Wicker (DMNS) is thanked for the amazing photographs of the specimen used in figures 2, 8,
and 9. Lindsay Dougan (Digital Research Laboratory, Denver Museum of Nature & Science)
is acknowledged for initial segmentation and Matthew Colbert (University of Texas High-
Resolution X-ray Computed Tomography Facility) for scanning. The mesmerizing artwork
provided in figure 11 was created by Andrey Atuchin. We finally would like to thank all
individuals of the 2015 Mahajanga Basin field crew involved in the collecting of this
specimen.

**References**

Albrecht PW. 1976 The cranial arteries of turtles and their evolutionary significance. *J.*
*Morphol.* **149**, 159–182.
Alcalde L, Derocco NN, Rosset SD. 2010 Feeding in syntopy: diet of *Hydromedusa tectifera*
and *Phrynops hilarii* (Chelidae). *Chel. Cons. Biol.* **9**, 33–44.
Ali JR, Krause DW. 2011 Late Cretaceous bio-connections between Indo-Madagascar and
Antarctica: refutation of the Gunnerus Ridge causeway hypothesis. *J. Biogeogr.* **38**, 1855–
1872.
Anderson NJ. 2009 Biomechanics of feeding and neck motion in the softshell turtle, *Apalone*
*spinifera*, Rafinesque. Ph.D. Thesis, Department of Biology, Idaho State University.
Anquetin J, Tong H, Claude J. 2017 A Jurassic stem pleurodire sheds light on the functional
origin of neck retraction in turtles. *Sci. Rep.* **7**, 42376.
Batsch AJGC. 1788 Versuch einer Anleitung zur Kenntniss und Geschichte der Thiere und
Mineralien. Erster Theil. Allgemeine Geschichte der Natur; besondre der Säugthiere,
Vögel, Amphibien und Fische. Jena: Akademische Buchhandlung.
Benson RBJ, Starmer-Jones E, Close RA, Walsh SA. 2017 Comparative analysis of
vestibular ecomorphology of birds. *J. Anat.* **231**, 990–1018.
Bona P, de la Fuente MS. 2005 Phylogenetic and paleobiogeographic implications of
*Yaminuechelys maior* (Staesche, 1929) new comb., a large long-necked chelid turtle from
the early Paleocene of Patagonia, Argentina. *J. Vert. Paleontol.* **25**, 569–582.
Bourque JR. 2016 Side-necked turtles (Testudines, Pleurodira) from the ancient Gulf coastal
plain of Florida during middle Cenozoic megathermals. *Chelon. Conserv. Biol.* **15**, 23–35.
Bramble DM, Wake DB. 1985 Feeding mechanisms of lower vertebrates. In Functional
Vertebrate Morphology (eds. M Hildebrand, DM Bramble, KF Liem, DB Wake). pp 230–
261. Cambridge, Harvard University Press.

Broin F de. 1988 Les tortues et le Gondwana. Examen des rapports entre le fractionnement
du Gondwana et la dispersion géographique des tortues pleurodires à partir du Crétacé.
*Stud. Palaeocheloniol.* **2**, 103–142.
Bronzati M, Benson RBJ, Evers SW, Ezcurra M, Cabreira S, Choiniere J, Dollman K,
Paulina-Carabajal A, Radermacher V, da Silva L, Sobral G, Stocker M, Witmer L, Langer
12 M, Nesbitt S. Deep evolutionary diversification of archosaur locomotion reflected by
13 vestibular morphology. *Curr. Biol.* in review.
Cadena E-A, Scheyer TM, Carrillo-Briceño JD, Sánchez R, Aguilera-Socorro OA, Vanegas
20 A, Pardo M, Hansen DM, Sánchez-Villagra MR. 2020 The anatomy, paleobiology, and
21 evolutionary relationships of the largest extinct side-necked turtle. *Sci. Adv.* **6**, eaay4593.
Cope ED. 1865 Third contribution to the herpetology of tropical America. *Proc. Acad. Nat.*
*Sci. Phil.* **17**, 185–198.
Ernst CH, Barbour RW. 1989 *Turtles of the World*. Washington D.C.: Smithsonian
Institution Press.
Evans SE, Jones MEH, Groenke JR, Turner AH, Krause DW. 2014 New material of
*Beelzebufo*, a hyperossified frog (Amphibia: Anura) from the Late Cretaceous of
Madagascar. *PLoS One* **9**, e87236.
Evers SWE, Benson RBJ. 2019 A new phylogenetic hypothesis of turtles with implications
for the timing and number of evolutionary transitions to marine lifestyles in turtles.
*Palaeontol.* **62**, 93–134.
Evers SWE, Joyce WGJ. 2020 A re-description of *Sandownia harrisi* (Testudinata:
Sandownidae) from the Aptian of the Isle of Wight based on computed tomography scans.
*R. Soc. Open Sci.* **7**, 191936.
Evers SW, Neenan JM, Ferreira GS, Werneburg I, Barrett PM, Benson RBJ. 2019
Neurovascular anatomy of the protostegid turtle *Rhinochelys pulchriceps* and comparisons

of membranous and endosseous labyrinth shape in an extant turtle. *Zool. J. Linn. Soc.* **187**,
800–828.
Evers SWE, Rollot Y, Joyce WGJ. 2020 Cranial osteology of the Early Cretaceous turtle
*Pleurosternon bullockii* (Paracryptodira: Pleurosternidae). *PeerJ* **8**, e9454.
Fachin Teran A, Vogt RC, de Fatima Soares Gomez M. 1995 Food habits of an assemblage
of five species of turtles in the Rio Guapore, Rondonia, Brazil. *J. Herpetol.* **29**, 536–547.
Ferreira GS, Bandyopadhyay S, Joyce WG. 2018c A taxonomic reassessment of *Piramys*
*auffenbergi*, a neglected turtle from the late Miocene of Piram Island, Gujarat, India.
*PeerJ* **6**, e5938.
Ferreira GS, Bronzati M, Langer MC, Sterli J. 2018a Phylogeny, biogeography and
diversification patterns of side-necked turtles (Testudines: Pleurodira). *R. Soc. Open Sci.*
**5**, 171773.
Ferreira GS, Iori FV, Hermanson G, Langer MC. 2018b New turtle remains from the Late
Cretaceous of Monte Alto-SP, Brazil, including cranial osteology, neuroanatomy and
phylogenetic position of a new taxon. *Pal. Z.* **92**, 481–498.
Ferreira GS, Rincón AD, Solórzano A, Langer MC. 2015 The last marine pelomedusoids
(Testudines: Pleurodira): a new species of *Bairdemys* and the paleoecology of
*Stereogenyina*. *PeerJ* **3**, e1063.
Forster CA, O'Connor PM, Chiappe LM, Turner AH. 2020 The osteology of the Late
Cretaceous paravian *Rahonavis ostromi* from Madagascar. *Palaeontol. Electron.* **23**, a29.
Forster CA, Sampson SD, Chiappe LM, Krause DW. 1998 The theropod ancestry of birds:
new evidence from the Late Cretaceous of Madagascar. *Science*, **279**, 1915–1919.
Foth C, Rabi M, Joyce WG. 2017 Skull shape variation in extant and extinct Testudinata and
its relation to feeding and habitat. *Act. Zool.* **98**, 310–325.

Foth C, Evers SW, Joyce WG, Volpato VS, Benson RBJ. 2019 Comparative analysis of the
shape and size of the middle ear cavity of turtles reveals no correlation with habitat
ecology. *J. Anat.* **235**, 1078–1097.
Gaffney ES. 1979 Comparative cranial morphology of recent and fossil turtles. *Bull. Am.*
*Mus. Nat. Hist.* **164**, 65–376.
Gaffney ES, Forster CA. 2003 Side-necked turtle lower jaws (Podocnemididae,
Bothremydidae) from the Late Cretaceous Maevarano Formation of Madagascar. *Am.*
*Mus. Novitates* **3397**, 1–13.
Gaffney ES, Krause DW. 2011 *Sokatra*, a new side-necked turtle (Late Cretaceous,
Madagascar) and the diversification of the main groups of Pelomedusoides. *Am. Mus.*
*Novitates* **3728**, 1–28.
Gaffney ES, Tong H, Meylan PA. 2006 Evolution of the side-necked turtles: the families
Bothremydidae, Euraxemydidae, and Araripemydidae. *Bull. Am. Mus. Nat. Hist.* **300**, 1–
Gaffney ES, Krause DW, Zalmout IS. 2009 *Kinkonychelys*, a new side-necked turtle
(Pelomedusoides: Bothremydidae) from the Late Cretaceous of Madagascar. *Am. Mus.*
*Novitates* **3662**, 1–25.
Gaffney ES, Meylan PA, Wood RC, Simons E, de Almeida Campos D. 2011 Evolution of the
side-necked turtles: the family Podocnemididae. *Bull. Am. Mus. Nat. Hist.* **350**, 1–237.

vertebrates (eds. JHM Thewissen, S. Nummela), pp. 133–256. Berkeley: University of
California Press.
Goloboff PA, Farris JS, Nixon K. 2008 TNT. A free program for phylogenetic analysis.
*Cladistics* **24**, 774–786.
Goloboff PA, Torres A, Arias JS. 2018 Weighted parsimony outperforms other methods of
phylogenetic inference under models appropriate for morphology. *Cladistics* **34**, 407–437.
Hermanson G, Iori FV, Evers SW, Langer MC, Ferreira GS. 2020 A small podocnemidoid
(Pleurodira, Pelomedusoides) from the Late Cretaceous of Brazil, and the innervation and
carotid circulation of side-necked turtles. *Pap. Palaeontol.* **6**, 329–347.
Herrel A, O'Reilly JC, Richmond AM. 2002 Evolution of bite performance in turtles. *J. Evol.*
*Biol.* **15**, 1083–1094.
Joyce WG, Bandyopadhyay S. 2020 A revision of the pelomedusoid turtle *Jainemys*
*pisdurensis* from the Late Cretaceous (Maastrichtian) Lameta Formation of India. *PeerJ* **8**,
e9330.
Joyce WG, Anquetin J, Cadena E-A, Claude J, Danilov IG, Evers SW, Ferreira GS, Gentry
AD, Georgalis GL, Lyson TR, Pérez-García A, Rabi M, Sterli J, Vitek N, Parham JF.
2021 A nomenclature for fossil and living turtles using phylogenetically defined clade
names. *Swiss J. Paleontol.*
Joyce WG, Lyson TR, Kirkland JI. 2016a An early bothremydid (Testudines, Pleurodira)
from the Late Cretaceous (Cenomanian) of Utah, North America. *PeerJ* **4**, e2502.
Joyce WG, Rabi M, Clark JM, Xu X. 2016 A toothed turtle from the Late Jurassic of China
and the global biogeographic history of turtles. *BMC Evol. Biol.* **16**, 236.
Joyce WG, Parham JF, Lyson TR, Warnock RCM, Donoghue PCJ. 2013a A divergence
dating analysis of turtles using fossil calibrations: an example of best practices. *J.*
*Paleontol.* **87**, 612–634.

Joyce WG, Werneburg I, Lyson TR. 2013. The hooked element in the pes of turtles
(Testudines): a global approach to exploring primary and secondary homology. *J. Anat.*
**223**, 421–441.
Joyce WG, Rabi M, Clark JM, Xu X. 2016 A toothed turtle from the Late Jurassic of China
and the global biogeographic history of turtles. *BMC Evol. Biol.* **16**, 236.
Kennett RM, Tory O. 1996 Diet of two species of freshwater turtle from tropical northern
Australia. *Copeia* **1996**, 409–419.
Krause DW, Kley NJ. 2010 *Simosuchus clarki* (Crocodyliformes: Notosuchia) from the Late
Cretaceous of Madagascar. *J. Vertebr. Paleontol.* **30 (6, Supp.)**, 1–236.
Krause DW, Hoffmann S, Wible JR, Kirk EC, Schultz JA, v Koenigswald W, Groenke JR,
Rossie JB, O'Connor PM, Seiffert ER, Dumont ER, Holloway WL, Rogers RR,
Rahantarisoa LJ, Kemp AD, Andriamialison H. 2014 First cranial remains of
gondwanatherian mammal reveal remarkable mosaicism. *Nature* **515**, 512–517.
Krause DW, Sertich JJW, O'Connor PM, Curry Rogers KA, Rogers RR. 2019 The Mesozoic
biogeographic history of Gondwanan terrestrial vertebrates: insights from Madagascar's
fossil record. *Ann. Rev. Earth Planet. Sci.* **47**, 519–553.
Krause DW, Hoffmann S, Hu Y, Wible JR, Rougier GW, Kirk EC, Groenke JR, Rogers RR,
Rossie JB, Schultz JA, Evans AR, von Koenigswald W, Rahantarisoa LJ. 2020a Skeleton
of Cretaceous gondwanatherian mammal from Madagascar reflects long-term insularity.
*Nature* **581**, 421–427.
Krause DW, Groenke JR, Hoffmann S, Rogers RR, Rahantarisoa LJ. 2020b Introduction to
*Adalatherium hui* (Mammalia, Gondwanatheria) from the Late Cretaceous of Madagascar.
*Soc. Vert. Pal. Mem.* **21**, 4–18.
Krause DW, Evans SE, Gao K-Q. 2003 First definitive record of Mesozoic lizards from
Madagascar. *J. Vert. Paleontol.* **23**, 842–856.

Kummer S, Heiss E, Singer K, Lemell P, Natchev N. 2017 Feeding behaviour and feeding
motorics in subadult European pond turtles, *Emys orbicularis* (Linnaeus, 1758). *Act. Zool.*
*Bulg. Suppl.* **10**, 77–84.
Lakjer T. 1926 Studien über die Trigemini-versorgte Kaumuskulatur der Sauropsiden. CA
Reitzel, Copenhagen, Denmark

[revised manuscript text omitted]

Appendix B

Dear Editors,

Many thanks for providing us with the prompt reviews of our manuscript regarding a new species of fossil turtle from the Late Cretaceous of Madagascar.

We were pleased to see that all three reviewers agreed with the primary conclusions of our study. All reviewers also provided us with extremely minor comments that we were able to address with ease. Our point-by-point responses are below.

Our final submission consists of a clean manuscript file, a manuscript file with tracked changes, 11 figures in PDF format, and five supplementary files.

Please feel free to let us know if any issues remain. We otherwise look forward to seeing this manuscript in print.

With best regards,

Walter Joyce

Responses to Reviewers:

Reviewer 1: The paper presents a new species of fossil turtle based on excellent fossil material from the Cretaceous of Madagascar. The paper includes a full description of the material, which is professional and carefully documented, and the analytical part includes a phylogenetic analysis, which again is professional and excellent, and this leads to consequences on evolution of feeding modes among pleurodiran turtles. The author team includes world-leading experts on fossil turtles, turtle phylogeny, the Madagascar Cretaceous faunas, and interpretation of these fossils. The authors establish a new family, a new genus, and a new species.

Response: *Many thanks for the positive comments.*

Reviewer 1: Pages 4, 5: iron out repetitions about Maevarano Formation (Maastrichtian age; faunal list) between Introduction and Geological setting.

Response: *To avoid the appearance of repetition, we removed the section regarding the Maevarano Formation from the Introduction and reworked the relevant information into the Geological Setting section.*

Reviewer 1: Page 5: Geological setting [not settings]

Response: *OK, fixed.*

Reviewer 1: Page 10, line 6: Sahnachelyidae [not italic]; also check wrongly italicized higher taxon names at lines 26, 58, page 11, lines 3, 14, 15, 17, 28, 29, etc... page 47, line 3] or is it now appropriate to italicize these?]

The figures are lovely, especially the final, reconstruction, figure.

Response: We highlight in section "3.2. Nomenclature" that all higher taxonomic names are clade names that are placed in italics following the rules of the newly established PhyloCode. We quickly checked for consistency, but everything looks to be fine. We also thank the reviewer for the compliment about the figures, which we will gladly forward to the artist.

Reviewer 2: In my opinion, this is an interesting and well-written paper. Therefore, I recommend its publication in this journal.

Response: Thanks for the positive comments.

Reviewer 2: I indicate here some minor suggestions:

- I recommend the authors to add a diagnosis for the new clade Sahnachelyidae. Some or all of the characters that could diagnose this clade are indicated throughout the Discussion section, but they should be grouped, as a diagnosis, in the Systematic paleontology section.

Response: We provide a differential diagnosis for our new species that includes all higher taxonomic units, including Pleurodira, Pelomedusoides, and Sahnachelyidae. Although it would be possible to break this apart, that would rupture the flow. As other readers may also wonder if there are diagnostic characters for the clade Sahnachelyidae, we added a sentence to the diagnosis of the species that highlights the relevant characters as such and a section to the clade Sahnachelyidae that refers the reader to the species diagnosis. This should get the job done as well.

Reviewer 2: The order of the figures is not the same as that in which they are cited: Figure 10 is one of the first to be referred.

Response: We switched the order of the figures to reflect their order of first appearance.

Reviewer 2: I recommend including all the comparisons between the new species and its sister taxon Sokatra in the Discussion section but not in the Description section (and, to avoid repetitions, remove all of them from the Description section).

Response: This issue is a question of preference. As is, comparisons are summarized in the diagnosis, but otherwise interspersed in the description, which is quite useful as well. So, unless the editors object, we decided to keep things as is.

Reviewer 2: I suggest removing the reference to "Bronzati et al. in review": Unaccepted or unpublished papers should not be referred.

Response: We updated this reference to be "in press".

Reviewer 2: A minor mistake is detected in Figure 8: neural VI is indicated as IV. A recommendation in that figure: the abbreviations of all the scutes are written with only the initial letter in capital letters, except those for GU and EG. Could this be modified for those two scutes?

Response: *Many thanks for paying such close attention. We gladly addressed these two points.*

Reviewer 3: I made comments directly on the manuscript. They are highlighted with yellow text marker and comments. the most relevant suggestions/comments are:

Response: *Many thanks for critically reading the full manuscript. We gladly addressed the majority of suggested modifications highlighted in the PDF.*

Reviewer 3: 1) I think it is better to suggest a new species name included in the already established genus Sokatra. The decision of creating a new genus and species is arbitrary and there is no direct evidence for it. Sokatra is a monotypic genus and including a second species found as sister to the type species (Sokatra antitra) is a better taxonomic decision in my opinion. See further comments along the text.

Response: *The question whether we should place our new species in an existing or new genus is legitimate, but not fruitful, as it simply is a matter of choice (as noted by the reviewer). This is the crux of taxonomy! Although we could add a brief section outlining why we think naming a new genus is reasonable, for instance by highlighting that the amount of variation we see between our new species with Sokatra antitra is beyond levels seen in extant genera, we would rather not engage in this debate here. So, given that the other two reviewers did not object, we decided to keep this as is and hope that the editors find that decision acceptable.*

Reviewer 3: Neck measurements M&M and results need to be reviewed.

2.1) Imprecision: The most inaccurate/imprecise would be measuring with ImageJ and caliper and mixing different measurements from different measuring methods. This is not the case and the error is minimized. See Mariani & Romano, 2017 DOI 10.7717/peerj.2890. All measurements were performed by the same author? Better, to avoid different individual error input. I recommend to take the same measurements more than one time (x3?) and use mean values as entries. An ANOVA can be done to detect differences and measure the error. The statement that this measurements "should be seen as approximations" is confusing. Should I trust the results? If you present a description of the measurements "errors" (basic statistics, including standard deviation) I can check if the data is ok or not.

2.2) Contradiction: In M&M it is stated that "missing neck vertebrae was estimated based on averages of the preserved vertebrae" (see page 9 of the manuscript). In results (see page 46 of the manuscript) it is stated that it was assumed (for the new species) "... the remaining [i.e., excluding atlas and axis] six cervicals had similar dimensions, we calculated a neck length of approximately 14cm." Thus, the missing entries were estimated differently in all

species used for comparisons? Therefore, different assumptions were made and this is reckless. If I understand this correctly, assuming C3-C8 equal midline length of C2 (axis) means that C2-C8 has 126mm (18mm x7). Thus, C1 (atlas) has 14mm (the measure was not informed by the authors). If we use the "averages of the preserved vertebrae" C3-C8 should be estimated as 16mm each, leading to a neck of ~128mm and a neck to carapace ratio ~50%. This is closer to *Chelonoidis denticulata* (48%), *Pelomedusa subrufa* (49%), *Cuora amboinensis* (51%), and *Geoclemys hamiltonii* (51%). Besides that, C3 is partially preserved and according to figure 9b, I would say it is possibly shorter than axis (C2). The atlas (C1) is clearly shorter than 14mm. Ultimately, the morphometrics need to be revised and these estimates for UA 10581 are not reliable.

2.3) Persuasion: In the same paragraph at page 46, the authors say "... gives a neck to carapace ratio of about 55%, which is comparable to the ratio of long necked chelids, particularly *Chelys fimbriatus* (59%...". However, it is even closer to *Terrapene carolina* (53%), *Kinixys belliana* (54%), *Sacalia quadriocellata* (55%), *Sternotherus depressus* (56%), and *Kinosternon baurii* (57%). The direct comparison to *matamata* is misleading to reader and the authors are ignoring their own data to inform a precise comparison. This seems to be skewed to provide stronger support to their conclusions of parallel evolution. It is not necessary because the qualitative description and explanation about putative adaptation to suction feeding in this new species is well written and funded in the literature.

Response: *The neck ratios we utilize to inform our discussion regarding the possible paleoecology of our new species of extinct turtle are not the focus of our paper, but rather accessory information. Although taking measurements directly from specimens would have been preferential, this was not feasible due to the ongoing coronavirus restrictions. However, even if such restrictions were not in place, taking such measurements from specimens would not have been a simple feat, as much travel would have been needed to assemble a good dataset. As the relative length of the neck of turtles varies drastically across the tree, we chose to evaluate this feature by reference to photographs of museum specimens to which we have access. The fact that this material only yields a "horse backs estimate" is not problematic as we only interpret the data in extremely broad terms and because we are upfront about the fact that we are only providing approximations. The reviewer's proposal to assess precision by taking multiple measurements and by subjecting them to an ANOVA as first sight seems reasonable, but we reject this approach, as this would only assess the precision of the measurements we took from the photographs, not from the objects themselves. This is beside the point and would yield a false sense of precision. It must also be emphasized that, owing to incompleteness, deformation, and fracturing of fossils, measurements on all except the most pristine specimens must be considered imprecise.*

That being said, the fact that the reviewer spent much time on this aspect of our study suggests that we did not sufficiently highlight the coarse nature of our data. We therefore implemented the following changes:

** We clarify in the M&M that all measurements were taken by the same person.*

** We expanded the following sentence in the M&M*

“Given the imprecision associated with this methodology, all measurements should be seen as approximations **that possibly deviate from the true measurement by a few percentage points.**”

* *The reviewer was right to admonish apparent inconsistencies in the way we estimated missing vertebrae for fossil turtles, but this is due to poor description of the methods, not actual inconsistencies. We therefore rephrase the relevant sentence as:*

“The length of missing neck vertebrae for fossil turtles was either estimated by reference to the length of preserved neighboring vertebrae, excluding cervical I (the atlas), or by averaging and then extrapolating the length of preserved cervicals to the full column, again to the exclusion of cervical I. In either case, cervical I was disregarded, as it typically is much shorter than all subsequent vertebrae. If missing, the length of cervical I was estimated at one half of the length of cervical II. Given the numerous imprecisions associated with this methodology, all measurements and the resulting ratios should be seen as approximations that possibly deviate from the true measurement by a few percentage points.”

* *We updated our estimate of the neck length of our new species as 13.6 cm (we initially had rounded it to 14 cm) and the relative neck length as 53%.*

* *Although we are indeed most interested in comparisons with the extreme suction feeder *Chelus fimbriatus*, we do not wish to raise suspicions that we are neglecting other comparisons. We therefore expanded the following sentence:*

“This length, combined with the midline length of the carapace (25.5cm), gives a neck to carapace ratio of about 55%, which, keeping the imprecision of our measurement in mind, is broadly comparable to that of the box turtle *Terrapene carolina* (53%), the alligator snapping turtle *Macrochelys temminckii* (53%), the tortoise *Kinixys belliana* (54%), and only slightly shorter than the "long necked chelid" *Chelus fimbriatus* (59%, see electronic supplementary material).”

* Last, but not least, we remind the reader in every section dealing with these data that we are only providing rough estimates that should be viewed with some caution.

Reviewer 3: Phylogeny. 3.1) I was able to replicate this search using TNT v1.5. However, some small details could be provided. For instance, I found 29 MPTs of 55.60618. I used random seed as 1 and saving only 10 trees per replica; therefore, using default setting given that these configurations were not mentioned in M&M. I found the same topology strict consensus (provided as suppl material). After increasing the search strategies settings, by using random seed zero, saving up to 200 trees in each replica and collapsing trees after search, I found 33 MPTs. It is important to notice that the general memory should be adjusted to more than 100 trees (TNT default) in order to achieve these results. I've also run an analysis assuming a priori equal weights (=1) to all characters. This resulted on 5400 MPTs of 1295 steps. The topology is different (specially not recovering Chelidae as a clade), but there are several similarities to tree found in the original analysis. It is important to notice that the new species is found as sister to Sokatra antitra and that this clade is nested in a polytomy with *Atolchelys lepida* + *Podocnemidoidea*+*Bothremydidae* clade. Thus, the

assumption about weighting of characters do not change the general structure of the preferred phylogeny (fig 10).

Response: *We are relieved to see that the reviewer was able to replicate our results, but he is right to highlight that we failed to provide some information. We therefore rewrote the section as follows:*

“The matrix was subjected to a parsimony analysis using the traditional search options of the software TNT (Goloboff et al., 2008). **Unless stated below, all default settings of the program were maintained.** All characters that form morphoclines were ordered and are noted as such in the matrix, **in particular characters 1, 10, 14, 18, 19, 51, 52, 56, 57, 75, 78, 81, 82, 86, 88, 95, 96, 99, 101, 103, 112, 114, 115, 119, 128, 130, 171, 172, 174, 175, 182, 183, 193, 195, 202, 220, 224, 225, 231, and 242 (counting from 1, not 0).**

Proganochelys quenstedti was selected as the outgroup. **The memory was expanded to 10,000 trees.** Light implied weighting was implemented with a k value of 12 (Goloboff et al., 2018), and 1,000 replicates of random addition sequences were followed by a round of tree bisection and reconnection.”

Reviewer 3: 3.2) *Caninemys tridentata* is missing in the full strict consensus topology provided as supplementary material. BTW, not following Cadena, who considered *Caninemys tridentata* as junior synonym of *Stupendemys geographicus*?

Response: *This is an oversight that we gladly fixed quickly by combining *Caninemys tridentata* and *Stupendemys geographicus* into a single terminal. Although the revised matrix only retrieves 11 trees, the topology remains the same for our area of interest. We nevertheless made sure to correctly update the manuscript and all affected electronic supplementary materials.*

Reviewer 3: 3.3) There is an "alien" topology in the supplement material provided: TNT file with the character/taxon matrix. Please, check it. This is in the "tread block" at the end of the file, right after the list of multistate characters that should be treated as additive (ccode block). I'm attaching this topology, that is the following (parenthetical):

```
((((((23 (22 21)) (76 (77 (72 (69 (73 71 70 (75 74))))))) (((3 (6 (4 5))) 12) ((16 15) (((14 13) (8 7)) (11 (10 9)))))) 1) ((((((25 24) (27 26)) (17 (20 19))) 2) 78) ((28 18) (33 ((30 31 (37 ((34 (36 35)) ((40 38 39) (43 (42 (41 ((49 48) ((47 46) (45 44))))))) ((60 59) (53 (54 (55 ((58 (56 57)) (52 (51 50)))))))))) (61 (62 (63 (65 (64 ((66 (67 68)) (79 ((80 81 83 82) (86 89 84 87 88 85 (90 (95 (96 (98 ((91 (92 (103 93 94)) (102 100 101 (99 97))))))))))))))))))
```

Response: *We removed the block, thanks.*

Reviewer 3: That being said, it is important to highlight the quality of the manuscript. The description of the new species is presented in a very detailed and careful way and very well illustrated. The hypothesis of convergent evolution of suction feeding is supported by clear evidence and convincing. All abovementioned comments are minor details that I hope can improve even more the quality of the manuscript. Congratulations to the authors for this contribution.

Response: Thanks for the positive comments, but also the constructively critical insights!